# Rapid 3D phenotypic analysis of neurons and organoids using data-driven cell segmentation-free machine learning

Philipp Mergenthaler[1,2,3☯]*, Santosh Hariharan[1,4☯], James M. Pemberton[1,4], Corey Lourenco[4,5], Linda Z. Penn[4,5], David W. Andrews[1,4]*

**1** Biological Sciences, Sunnybrook Research Institute, University of Toronto, Toronto, Ontario, Canada, **2** Charité — Universitätsmedizin Berlin, Department of Experimental Neurology, Department of Neurology, Center for Stroke Research Berlin, NeuroCure Clinical Research Center, Berlin, Germany, **3** Berlin Institute of Health (BIH), Berlin, Germany, **4** Department of Medical Biophysics, University of Toronto, Toronto, Ontario, Canada, **5** Princess Margaret Cancer Centre, University Health Network, University of Toronto, Toronto, Ontario, Canada

☯ These authors contributed equally to this work.
* philipp.mergenthaler@charite.de (PM); david.andrews@sri.utoronto.ca (DWA)

**Data Availability Statement:** All 3D microscopy data sets generated in this study were deposited to the Open Microscopy Image Data Resource

## Abstract

Phenotypic profiling of large three-dimensional microscopy data sets has not been widely adopted due to the challenges posed by cell segmentation and feature selection. The computational demands of automated processing further limit analysis of hard-to-segment images such as of neurons and organoids. Here we describe a comprehensive shallow-learning framework for automated quantitative phenotyping of three-dimensional (3D) image data using unsupervised data-driven voxel-based feature learning, which enables computationally facile classification, clustering and advanced data visualization. We demonstrate the analysis potential on complex 3D images by investigating the phenotypic alterations of: neurons in response to apoptosis-inducing treatments and morphogenesis for oncogene-expressing human mammary gland acinar organoids. Our novel implementation of image analysis algorithms called Phindr3D allowed rapid implementation of data-driven voxel-based feature learning into 3D high content analysis (HCA) operations and constitutes a major practical advance as the computed assignments represent the biology while preserving the heterogeneity of the underlying data. Phindr3D is provided as Matlab code and as a stand-alone program (https://github.com/DWALab/Phindr3D).

## Author summary

Fluorescence microscopy is a fundamental technology for cell biology. However, unbiased quantitative phenotypic analysis of microscopy images of cells grown in 3D organoids or in dense culture conditions in large enough numbers to reach statistical clarity remains a fundamental challenge. Here, we report that using data-driven voxel-based features and machine learning it is possible to analyze complex 3D image data without compressing them to 2D, identifying individual cells or using computationally intensive deep learning

repository (http://idr.openmicroscopy.org) under accession number idr0105. The entire multicolor 3D confocal data set for the BH3 mimetics experiments (underlying Figs 3, 4, Fig A in S1 Text) is 2.74 TB in size and consists of 437400 single plane images (IDR "screen A"). The excitotoxicity data set (underlying Fig C in S1 Text) is 83 GB in size and consists of 33075 single plane images (IDR "screenB"). The MCF10A organoid data set (underlying Figs 5, 6, Figs D-F in S1 Text) is 15 GB in size and consists of 99786 single plane images (IDR "screenC"). Test image data sets with an excerpt of the neuron and organoid data that will allow quick evaluation of the core features of Phindr3D were deposited to the open science platform Zenodo (https://zenodo.org; neuron data: DOI 10.5281/zenodo.4064148 [https://dx.doi.org/10.5281/zenodo.4064148]; MCF10 organoid data: DOI 10.5281/zenodo.4384912 [https://dx.doi.org/10.5281/zenodo.4384912]). Numerical data for the figure panels were deposited to Zenodo (https://zenodo.org; DOI: 10.5281/zenodo.4385040 [https://dx.doi.org/10.5281/zenodo.4385040]) Images for Fig 2 and feature data for Fig 2B were provided by PerkinElmer, Hamburg, Germany, and used with permission; detailed description of the data can be found in the methods section and the original image data can be requested from alexander.schreiner@perkinelmer.com. Code availability: The Phindr3D method is available as Matlab 2017b code and as a compiled stand-alone executable software for Windows. It has been tested on 64-bit versions of Windows 7, Windows 10 and (virtual) Windows Server 2016. When running from Matlab, Phindr3D requires "Image Processing and Statistics" and "Machine Learning" toolboxes to be installed. The software and code is freely available for non-commercial use and is released under the GNU General Public License Version 3. It is accessible on GitHub (https://github.com/DWALab/Phindr3D).

**Funding:** This work was supported by CIHR Foundation grant FDN 143312 (DWA), CIHR grant PJT 156167 (LZP), the European Union's Seventh Framework Programme (FP7/2008–2013) under Grant Agreement 627951 (Marie Curie IOF to PM), the German Academic Exchange Service (DAAD) with funds from the German Federal Ministry of Education and Research (57212163 to PM), and in part by the Bundesministerium für Bildung und Forschung, Germany (BMBF, grant no. 16GW0191 to PM). JMP is recipient of the Queen Elizabeth II graduate scholarship in science and technology. PM has been supported by the BIH-Charité Clinical Scientist Program funded by the Charité – Universitätsmedizin Berlin and the Berlin Institute of Health. DWA holds a Tier 1 Canada Research

techniques. Further, we present methods for analyzing this data by classification or clustering. Together these techniques provide the means for facile discovery and interpretation of meaningful patterns in a high dimensional feature space without complex image processing and prior knowledge or assumptions about the feature space. Our method enables novel opportunities for rapid large-scale multivariate phenotypic microscopy image analysis in 3D using a standard desktop computer.

This is a *PLOS Computational Biology* Methods paper.

## Introduction

High content analysis (HCA) and phenotypic profiling together with genetic pathway analysis or compound screening have helped elucidate fundamental biological processes [1] and aid the discovery of novel therapeutic compounds [2]. A number of software tools for phenotypic profiling exist [3] but most require initial identification of specific cellular regions such as nuclei and cytoplasm by image segmentation [4,5]. Subsequent cell image analysis is typically performed by extracting large preconceived feature sets and followed by feature selection using automated procedures [6]. However, segmentation of complex structures can be inherently difficult, requires expert knowledge, and demands specific markers. Likewise, selection of meaningful phenotypic features is not straightforward. Analyses of three-dimensional (3D) organoid cultures or complex cellular phenotypes, such as in dense neuronal cultures, are difficult with conventional methods. While segmentation can be achieved for defined structures such as nuclei in spheroids [7] and volumetric tracing of single neuronal arbors in drosophila brains [8], large-scale analysis of complex, hard-to-segment 3D microscopy data is very hard to do with current tools. An attractive way to circumvent the problem of cell segmentation involves using whole image features [9,10]. However, these methods have not been applied to 3D images and standard implementations are not appropriate for micrographs of organoids as much of the 3D image volume is devoid of cells. Furthermore, analysis of organoid images in which image stacks are collected at different distances from the objective lens results in non-uniform focus within and between organoids. As a more practical method of analyzing micrographs without cell segmentation we implemented data driven feature calculation methods similar to those used for 2D image exploration in PhenoRipper [9], which are related to the concept of spatial pyramids in computer vision [11], and for image classification by various 'deep-learning' neural networks. We further advanced the method by implementing automated decision making by integrating algorithms for classification and clustering. Deep learning methods have now been successfully applied to various aspects of microscopy image analysis such as image restoration, classification, or segmentation [12]. However, a comprehensive architecture for high content analysis in 3D has not yet been established.

To implement automated decision making from 3D datasets we created and extensively validated automated methods for identifying image slices containing information rich data and discarding slices with low or misleading information content. We also validate a method for optimizing the number of clusters identified by affinity propagation. We have named our method Phindr3D. It comprises integrated computer vision, phenotyping, and analysis framework for 3D image analysis using data-driven features derived from complex 3D microscopy image data without object identification. We demonstrate here the utility of Phindr3D to study inhibition of anti-apoptotic proteins in dense cortical neurons and the effects of oncogenes on organoid morphologies, both datasets intractable by conventional methods of

Chair (CRC) in Membrane Biogenesis. LZP holds a Tier 1 CRC in Molecular Oncology. The funders had no role in study design, data collection and analysis, decision to publish, or preparation of the manuscript.

**Competing interests:** The authors have declared that no competing interests exist.

analysis. Phindr3D is provided as Matlab code and as a stand-alone program (https://github.com/DWALab/Phindr3D).

## Results

### Shallow learning of data-driven features in 3D microscopy images

In Phindr3D image histograms are calculated iteratively and categorized at different hierarchical levels to automate learning and computing features from the image data (Fig 1). This circumvents the need for cell segmentation and feature selection, is much less computationally intensive than deep-learning neural-networks, and performs well with as little as 10 images as training data. Therefore, we refer to it as "shallow learning".

Our method applies two steps to multichannel 3D image data: First, pixels are binned into pixel categories based on normalized pixel intensities using unsupervised k-means clustering. Second, neighborhood pixels in 3D are combined to create a supervoxel (SV) which is then described by the normalized frequency of pixel categories and retains the spatial association of pixels within the SV, thereby creating an SV image stack. These two steps are then repeated on

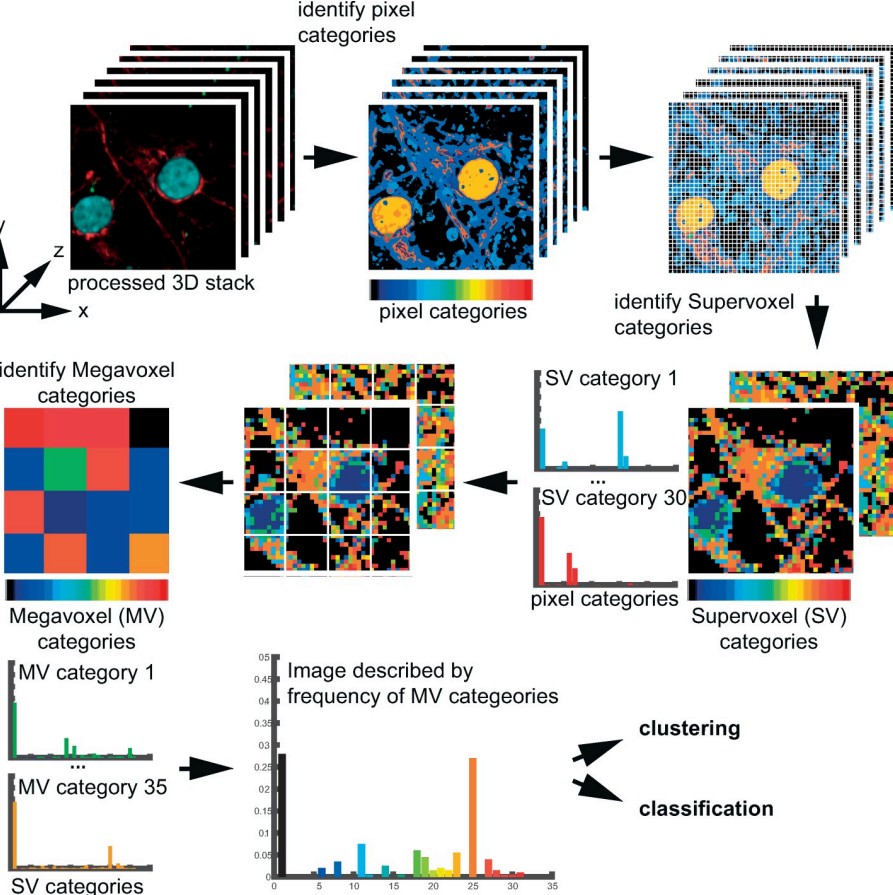

**Fig 1. Overview and validation of the Phindr3D method.** In a cell segmentation-independent method, Phindr3D iteratively performs two steps on 3D multichannel images. It calculates and categorizes image histograms at different hierarchical levels to compute data-driven image features using unsupervised clustering. Pixels are binned into pixel categories using k-means clustering and supervoxels (SV) are generated by combining pixels from neighboring z-slices. Next, SVs are combined into megavoxels (MV) and MVs are categorized. Each image stack is then defined by the normalized frequency of these different MV categories.

the SV image stack to generate megavoxels (MV) where each MV is described by the normalized frequency of the SV categories. Since the MVs are derived by combining neighborhood SVs, their spatial arrangement is still preserved. In Phindr3D, all these parameters can be adjusted by the user. Ultimately, an image is described by the normalized frequency and spatial hierarchy of these different MV categories. We provide visualization of the structure of the high-dimensional feature space in a 2D or 3D plot using both linear (principal component analysis, PCA) and non-linear (Sammon mapping; t-distributed stochastic neighbor embedding, t-SNE) dimension reduction [6]. While PCA and Sammon mapping preserve the global structure of the data, t-SNE performs non-linear mapping focusing on preserving local neighborhoods of data points. Integrating random forests classification and unsupervised affinity propagation (AP) clustering enables quantitative evaluation of query data. For visualization, the proportions of different groups defined by the metadata are represented with pie charts in cluster centers. This additional minor innovation enables visual confirmation that neighboring clusters are distinct while also illustrating the inherent heterogeneity of the image data.

## Data-driven 3D HCA of cancer cell lines after autophagic flux-arrest

For validation, we analyzed a HCA 3D reference data set consisting of three cell lines immunostained for the prototypic autophagic cargo protein SQSTM/p62 after autophagic flux-arresting treatment with increasing concentrations of chloroquine [13] (Fig 2). Importantly, the concentration response curves obtained using both Phindr3D and conventional methods resulted in very similar $EC_{50}$ values and dynamic range. This demonstrates that for relatively well-defined datasets cell segmentation-free data-driven feature calculation and classification resulted in accuracy comparable to supervised classification using curated and manually optimized feature sets (Fig 2). Thus, 3D HCA exploration by Phindr3D may be useful for automated rapid first line analysis of complex data sets even where cell segmentation is not difficult.

## Identification of distinct phenotypic responses in dense neuronal cultures upon pharmacological inhibition of anti-apoptotic proteins

The key feature of our method for cell segmentation-independent data-driven voxel-based learning of image stack features is its utility for analyzing hard-to-segment multichannel 3D microscopy data. This is especially relevant for optimal high-density neuronal cultures where accurate cell segmentation is not always possible and it is almost impossible to assign neuronal branches to individual cell bodies. Nevertheless, it is well known that high-density cultures more accurately reflect in vivo biology particularly for cell autonomous responses such as intrinsic apoptosis.

The Bcl-2 protein family is involved in the pathophysiological mechanisms contributing to neuronal apoptosis after stroke [14] or epileptic seizures [15]. Here, we used dense cultures of primary mouse brain cortical neurons to investigate the role of Bcl-2 proteins in neuronal survival. Neurons were treated with small molecule inhibitors of the anti-apoptotic proteins Bcl-2, Bcl-XL, and Bcl-w (BH3-mimetics) for 24 hours prior to staining and imaging (Figs 3 and 4, and Fig A in S1 Text). The entire data set consisting of 8100 3-channel image stacks (218700 individual TIF images) across 9 biological replicates in 3 plates (Fig 3) was processed using Phindr3D.

Phindr3D features are automatically derived from the data, therefore, to permit direct comparisons, Phindr3D features were calculated using pooled images from all three plates. After feature calculation, the data were analyzed both for individual plates separately (Fig 3B–3D) and in aggregate (Fig 4). The multivariate feature data for each drug concentration are shown

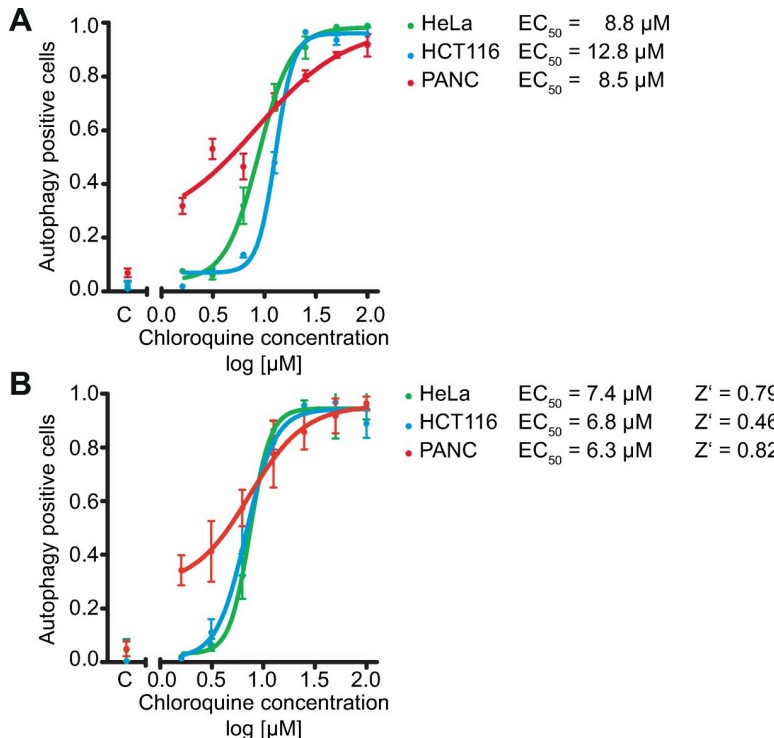

**Fig 2. 3D HCA of concentration response of cancer cell lines to autophagic flux-arresting chloroquine treatment.**
The analysis was based on immunofluorescence staining of the autophagy marker p62 (Autophagy positive cells).
Concentration response curves generated using classifiers for each cell line trained on control (C) and the highest
concentration for each. **A)** Result of the Phindr3D analysis. For the Phindr3D analysis, each original multichannel
multistack image was split into 16 equal parts to increase the number of data points for training in Phindr3D. The
classifier was built using Phindr3D features extracted from multichannel raw images. **B)** Replot of original analysis:
Supervised classification using a traditional cell segmentation-based algorithm from PerkinElmer. These data
including $EC_{50}$ and Z' values were provided by PerkinElmer and are shown here with permission for comparison of
the two analysis methods. Details on the generation of this data set have been published elsewhere [13]. Together, these
data demonstrate that data-driven cell segmentation-free phenotyping using the Phindr3D method can achieve similar
accuracy in 3D data to traditional cell segmentation-based supervised classification using curated feature sets.

using principal component analysis (PCA) for the three plate replicates, each of which contains
data for three biological replicates (Fig 3B–3D). Images of neurons treated with 30 nM concen-
trations mostly clustered with the 3 nM concentration of the same compound (Fig 3D, plate
1). Therefore, this concentration was excluded from plates 2 and 3, to enable exploration of
multiple concentrations of staurosporine (STS) and actinomycin D (ActD) with a similar plate
layout for all three experiments. Indicative of successful automated identification of subpopu-
lations, centroids for image stacks from different treatments occupied distinct regions in a
PCA plot (Fig 4A) with minimal variations between replicates (Fig A in S1 Text). Furthermore,
the locations of centroids for increasing treatment concentrations correspond with increasing
distances from the positions of the untreated and DMSO controls (Fig 4A, adjacent arrows).
Affinity propagation clustering with Phindr3D using a custom method for automated cluster
estimation by preference value tuning (Fig 5A and Fig B in S1 Text) revealed eight distinct
phenotype clusters for the combined neuronal data (Fig 4B). We used an angle-based method
for knee point detection for automatically determining the number of clusters [16]. Clustering
the data from each of the plates separately with the same features resulted in 6–7 clusters (Fig
3D). This was likely due to there being less data for the individual plate data sets resulting in a
lower signal to noise ratio obscuring some clusters. Nevertheless, the distribution of treatments

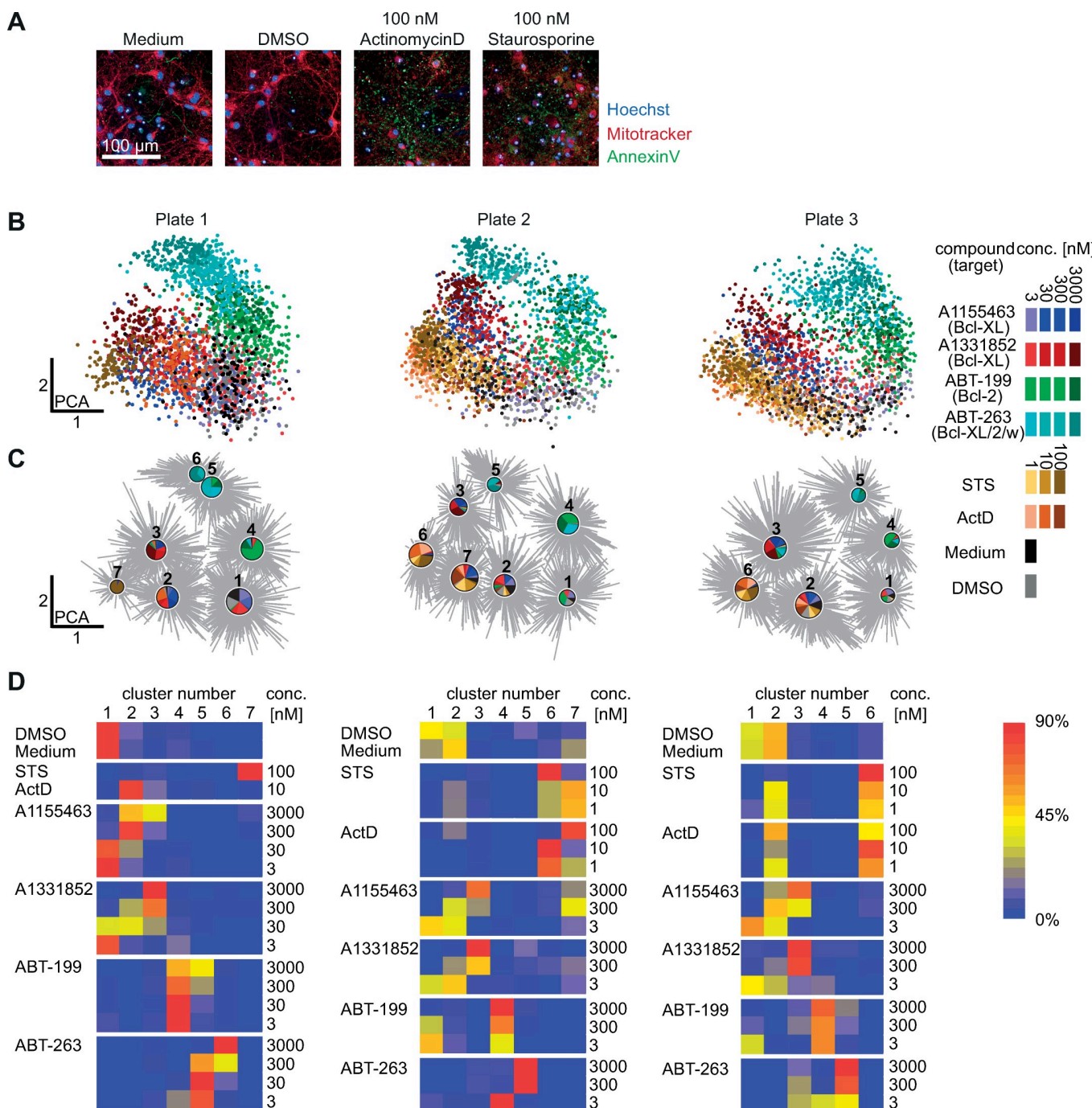

**Fig 3. Distinct phenotypic responses in dense neuronal cultures. A)** Representative images of neurons from control groups. **B)** Two dimensional principal component projection of 3D multichannel image feature space for each plate. Each point represents one 3D multichannel image. Only plate 1 contained the 30 nM concentration of the inhibitors of the anti-apoptotic proteins while plates 2 and 3 contain more concentrations for the STS and ActD controls (tan-brown dots, respectively) making the distributions appear more different than they actually are. **C)** Automated clustering using affinity propagation. The pie charts mark the cluster centroids and show the distribution of different treatments within each cluster. The sizes of the pie charts represent the number of data points within each cluster, cluster numbers are given above the pie charts. **D)** Heatmap of the distribution of phenotypes in response to the different treatments across clusters for each plate provide interpretative data complementary to but different from the pie charts in B) that show the distributions within clusters. *ActD–Actinomycin D*; *STS–Staurosporine*; *for B & C*: *the colors correspond to the different treatment groups as indicated in the legend for concentrations in nM.*

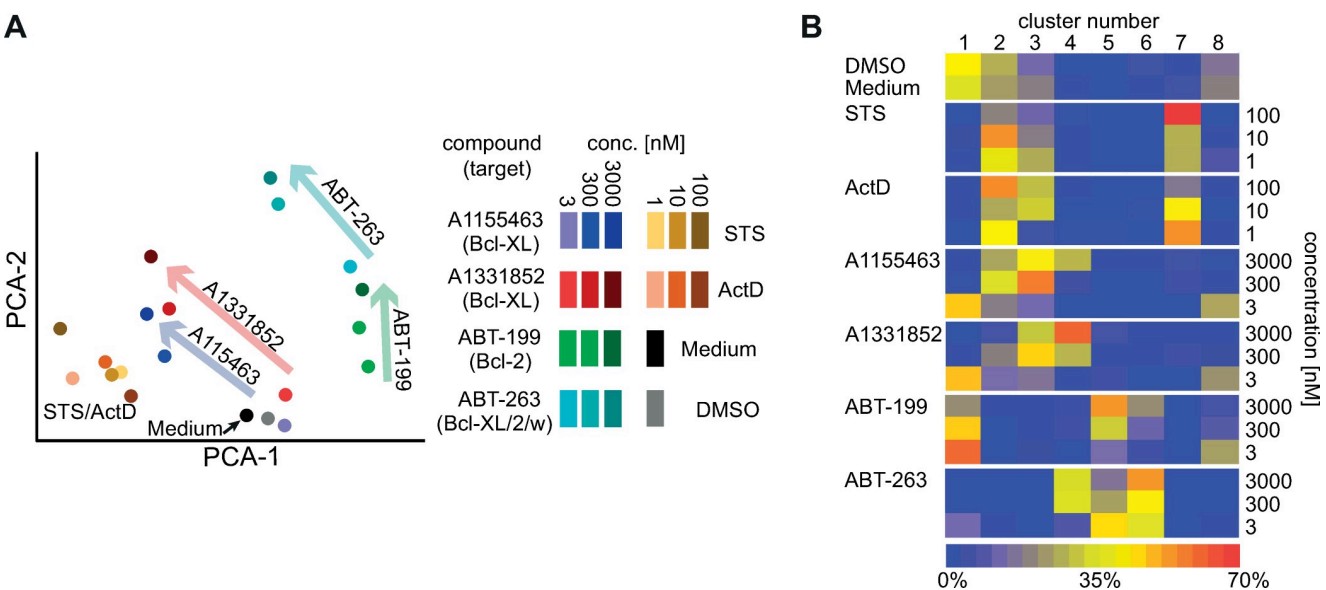

**Fig 4. Concentration response and aggregate multivariate phenotypic response of neurons treated with BH3-mimetics. A)** 2-dimensional PCA representation of the image feature space for neurons treated with BH3 mimetics (arrows, concentration increase) showing averaged treatment centroids colored by drug and concentration. **B)** Heatmap of the proportion of images of neurons treated with the indicated drugs assigned to individual clusters. *ActD–Actinomycin D, STS–Staurosporine; for A: the colors correspond to the different treatment groups as indicated in the legend for concentrations in nM.*

across the different clusters showed a similar profile for all replicates (Fig 3D). As expected, the two Bcl-XL inhibitors (Fig 3C, blue and red) clustered together. However, since all of the inhibitors of anti-apoptosis proteins trigger cell death by apoptosis it was unexpected that the different drugs resulted in well separated clusters. While the drugs have different potencies, the concentrations tested span at least three orders of magnitude therefore we anticipated the concentration response lines would intersect at concentrations that induced equivalent amounts of apoptosis. Nevertheless, the data for the Bcl-XL inhibitors was well separated from the specific Bcl-2 inhibitor ABT-199 (Fig 3C, green) and the inhibitor of Bcl-2, Bcl-XL and Bcl-w (ABT-263, Fig 3C, cyan) suggesting morphological differences in the responses of the cells to the drugs. However, the partial overlap between ABT-199 (green) and ABT-263 (cyan) is consistent with both drugs inhibiting Bcl-2 (Fig 3C). The lowest concentration of the Bcl-XL inhibitors (blue, red) clustered with the untreated cells (medium, black) and controls (DMSO, grey) suggesting minimal effects of this concentration on cell physiology. The mixing of drug concentrations within clusters likely reflects the heterogeneity in responses to the drugs (Fig 3C).

Heatmaps provide an alternative way to visualize the data. In this format it is clearer that the distribution of phenotypes for each treatment show that as expected the lowest concentrations of the Bcl-XL inhibitors had minimal effect on the cells as they resulted in a distribution of phenotypes similar to DMSO and untreated controls (Fig 3D, cluster 1) or to 1 nM ActD and STS (Fig 3D, cluster 2). However, some images of DMSO and untreated cells are also found in cluster 2. This reflects the heterogeneity within these cultures and that, as expected, there are also dying and/or stressed cells in the negative controls. As the concentration of the Bcl-XL inhibitors increases, the morphologies of the cells are affected commensurately and transition through clusters 1–3 (Fig 3D). These data further suggest that A1331852 is slightly more efficacious than A11555463.

Although the phenotypic heterogeneity is lost when the clustering data is reduced to plotting the centroids for each treatment in PCA space, this type of presentation reinforces further

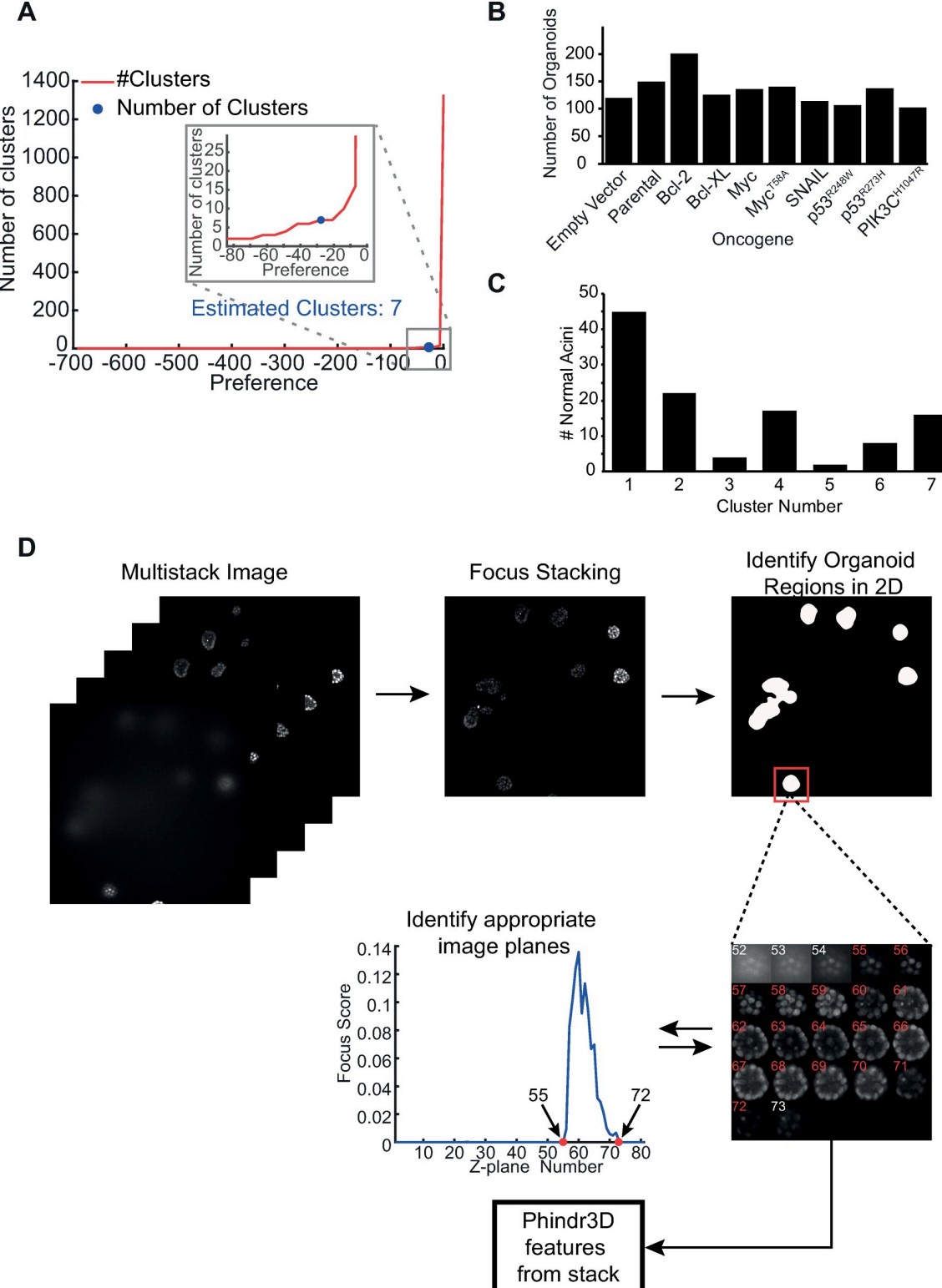

**Fig 5. Cluster estimation and workflow for organoid region identification from large image stacks. A)** Automated estimation of the preference value resulted in AP clustering generating seven clusters for the organoid dataset. A preference value that routinely results in a biologically meaningful number of clusters for micrographs of cells was automatically determined as the point at which the number of clusters begins to increase exponentially. This method differs from the standard method of setting the preference value in affinity propagation using median preference [17]. **B)** Total number of organoids of each type that were analyzed from images pooled from 3

replicate plates for MCF10A cells expressing each oncogene. All of the organoids that were well separated and at an appropriate height that sufficient planes were in focus were analyzed. At least 100 organoids were analyzed for each oncogene with a maximum number of less than 200. This level of variability in object number is unlikely to skew the clustering results. Consistent with this interpretation, organoids expressing Bcl-2 (with the largest number of organoids) and Bcl-XL (a protein of similar function with many less organoids) were found primarily in three clusters each, two of which were in common (Fig 6D). **C)** Identification of clusters containing normal acinar structures. Both parental and empty vector expressing cells resulted in heterogeneous organoid structures less than half of which were of normal morphology. This heterogeneity appears to be the result of spontaneous transformation of these premalignant cells. To determine which clusters were enriched in normal acinar structures we manually identified normal acinar structures by consensus for parental and empty vector samples. Manual identification was performed by inspection of 114 organoids by experts using only the nuclear channel. For each organoid top, middle and bottom z-planes were examined for roundness and a hollow interior. Then the distribution of normal mammary acinar organoids across clusters was determined. Cluster 1 contained both the largest number (45) of these normal organoids that account for 17% of the organoids in that cluster, therefore, it is the most likely cluster to represent normal structures. The next most representative of a relatively normal phenotype is cluster 2 containing 22 organoids, which account for 10% of the cluster. In contrast, cluster 7 contained 51% of the organoids expressing PIK3CA$^{H1047R}$ which account for 27% of the cluster and were also visually the most abnormal, suggesting this cluster represents an abnormal phenotype (Table A in S1 Text). **D)** Identification of organoids in a large image stack. First, multistack images of stained nuclei (DRAQ5) were used for organoid identification. Second, Focus Stacking was performed to flatten the image stack. For Focus Stacking, the pixel standard deviation in a 3 x 3 kernel was calculated for every pixel in the original image. Then for every pixel in the standard deviation image the gradient in an 11 x 11 pixels neighborhood was computed. The underlying assumption is that the higher the gradient magnitude for any pixel the better focused the image is. For every X-Y pixel location we identified the corresponding Z plane pixel with the highest gradient value. Finally, for each X-Y position in the image the original intensity of the pixel at the Z location of the best-focus was projected on a 2D plane to create a best-focus image that was used for further processing. Next, to identify organoid regions in 2D, we used contour segmentation by thresholding and watershed transformation. To identify the optically acceptable top and bottom planes for each organoid, a density curve from a normalized histogram of pixel value distribution within the organoid region in the best-focus image was calculated and the number of in focus pixels per Z-plane was used as a focus score (see graph). For n planes, the random chance of any plane being in focus is 1/n. Therefore, any Z-plane with a fraction of best-focus pixels more than 1/n was scored as sufficiently in focus for analysis. We set this value as the threshold and then arbitrarily added 2 more planes above and below the threshold. The numbers highlighted in red indicate the planes automatically chosen for a single organoid for 3D analysis from a gallery of Z-planes from the DRAQ5 channel for the single organoid shown to the right of the graph. The histogram shows the focus score for the 80 Z-planes recorded for the 2D location of the organoid. Red points in the histogram indicate the top and bottom planes automatically chosen for analysis (see Materials and Methods section for details).

some of the biology revealed by our data-driven feature analysis (Fig 4A). In this data display the differences in the responses of the cells to the different BH3 mimetics and the relationships between them is obvious. The concentration dependent differences between the Bcl-2 and Bcl-XL inhibitors are particularly striking.

As expected, in the averaged analyses neurons treated with low concentrations of the Bcl-2 and Bcl-XL inhibitors predominantly clustered together with control-treated (DMSO) neurons (Fig 4B). Furthermore, Bcl-2 inhibition and low concentration triple inhibition (Bcl-2/Bcl-XL/Bcl-w, cluster 5), high concentration triple (cluster 6) or Bcl-XL inhibition (clusters 3 and 4) formed clusters distinct from each other and from less-specific induction of cell death using actinomycin D or staurosporine (clusters 2 and 7). Altogether, the similarity between the two Bcl-XL inhibitors and separation between these and the other BH3-mimetics reinforced the unexpected observation that cultured neurons respond differently to the different classes of inhibitors even though they all trigger apoptosis. Visual re-examination of the images (Fig A in S1 Text) confirmed that the morphologies and staining patterns were qualitatively different. Moreover, even though individual data points are images containing many cells, PCA analysis of the Phindr3D data for individual images illustrate that our analysis based on data-driven features captured the heterogeneity in responses of cells to the drug treatments (Fig 3B and 3C and Fig A in S1 Text).

The specific inhibitors of Bcl-XL (A1155463 and A1331852) showed similar concentration trajectories with lower concentrations closer to and higher concentrations further away from DMSO and moving in the direction of higher concentrations of ActD and STS (Fig 4A). This result suggested that neurons in these cultures depend on Bcl-XL function for survival. In contrast, the averaged treatment centroids for the Bcl-2 inhibitor ABT-199 and the Bcl-XL/Bcl-2/Bcl-w triple inhibitor ABT-263 occupied space far away from DMSO and ActD/STS controls,

indicating different phenotypes for the different treatments. Nevertheless, with increasing concentration in this PCA representation the trajectory of the triple inhibitor ABT-263 was parallel to those of the Bcl-XL inhibitors yet displaced similarly to ABT-199 the inhibitor of Bcl-2 consistent with ABT-263 inhibiting both Bcl-XL and Bcl-2. Thus, the results of clustering may sensibly represent the known differences in the activities of the inhibitors.

To compute a concentration response using supervised classification, we generated a random forests classifier by training on data from DMSO negative controls, and as positive controls 10 nM ActD and 10 nM STS treatments (panel D in Fig A in S1 Text). At the higher concentrations, these drugs both kill cells albeit by different mechanisms. Therefore, to estimate cell death as a simple binary outcome due to the other compounds, the fraction of the images that were classified as either ActD or STS was computed. Unlike the results obtained by clustering, classification of these data (panel D in Fig A in S1 Text) demonstrated concentration-dependent effects on neuronal viability only for some drugs. With increasing concentrations of the Bcl-XL inhibitors, an increasing fraction of images was classified as either ActD or STS condition, suggesting a higher level of cell death with increasing concentrations of the Bcl-XL inhibitors. However, consistent with grossly normal brain development of Bcl-2 knockout mice [18], Phindr3D analysis of increasing concentrations of the Bcl-2 inhibitor ABT-199 did not result in increased classification as ActD/STS but clustering identified a distinct phenotype (Figs 3 and 4, and panel D of Fig A in S1 Text). Surprisingly, increasing concentrations of the triple inhibitor ABT-263 resulted in only a small trend towards increased classification as ActD/STS and clustered separately from all of the other treatments. This result was obtained despite the expected induction of neuronal cell death by this treatment. These results would suggest that triple inhibition of Bcl-XL, Bcl-2 and Bcl-w by ABT-263, resulted in phenotypes unlike the cell death induced by ActD/STS. Manual inspection of the corresponding 8100 3-channel image stacks is not feasible. However, the sample images (Fig A in S1 Text) suggest that neuronal morphologies are altered by ABT-199 and ABT-263. Classification did not detect cell death for Bcl-2 inhibition or low concentrations of triple inhibition (panel D in Fig A in S1 Text) because although the cells are altered, the phenotypes are too dissimilar from the controls (Fig 4A). These results highlight the utility of unsupervised data analysis for phenotypic profiling, particularly when investigating heterogeneous phenotypes or if the resulting phenotypes are unknown.

## Robust performance of data-driven Phindr3D image features

To measure robustness of the performance of the Phindr3D features, we collected 3D image stacks of three channel spinning disk confocal micrographs of neurons undergoing excitotoxic injury [14]. Phindr3D features were extracted from the image stacks using 40 MV categories and visualized in 2D by t-SNE projection of the 40-dimensional data (panel A in Fig C in S1 Text). The image dataset was then used to measure the effects on the performance of Phindr3D feature based analyses of added noise, blurring as well as effects of changing Phindr3D parameters. We discovered that although generally robust, the heterogeneity in the high frequency information in the images degraded the performance of Phindr3D. Consistent with this interpretation we found that adding random Gaussian noise initially improved and subsequently degraded performance (panel B,C in Fig C in S1 Text). Three-dimensional Gaussian blur (panel E,F in Fig C in S1 Text) also improved but had less dramatic effects on the already very good performance of the classifier. Adding high frequency noise effectively reduces the contribution of high frequency information in the images and therefore improves performance when this information is neither random nor informative. We assume this occurs when there is structured noise in the images resulting from the analysis of small cellular structures that do

not correlate with the phenotypes being investigated. Thus, for some datasets pre-processing the images may be useful. Finally, we used this image dataset and changed the Phindr3D parameters (panel F in Fig C in S1 Text) and found that classification of excitotoxicity is relatively insensitive to SV and MV sizes or number of SV categories. Nevertheless, if the SV and MV sizes were too large to capture relevant subcellular structure (i.e. 9x9x3), there was a detectable decrease which decreased even further at SV and MV sizes of 12x12x3. However, for other datasets we have found a much larger impact of SV and MV sizes, as they must be small enough for the phenotype of interest to affect SV categories sufficiently to provide discriminating power to the classifier. Thus, matching the Phindr3D parameters to the known biology of the system and the question being asked may be useful to optimize the results obtained. In summary, these data confirmed robust performance of our method to derive data-driven features from 3D image stacks. While dense cultures of cortical neurons cannot be usefully analyzed in 2D confocal images, the number of image planes in Z is relatively small and therefore does not compromise the speed of image collection compared to other structures like organoids.

## Oncogene-driven phenotypic alterations in human mammary gland acinar organoids

To further explore the utility of our Phindr3D method, we investigated the effect of oncogene expression (SNAIL, PIK3CA$^{H1047R}$, Myc, Myc$^{T58A}$, Bcl-2, Bcl-XL, p53$^{R248W}$, p53$^{R273H}$) compared to parental or empty vector (EV) on the phenotypes of 1330 acinar organoid cultures of human non-transformed MCF10A mammary gland epithelial cells stained for nuclei, mitochondria, and lysosomes (Figs 5B, 5C and 6, Fig D in S1 Text, Tables A and B in S1 Text). This is a particularly challenging dataset to analyze because the large size of the organoids results in optical aberrations that are non-uniform between image stacks, the image stacks are large and because the results of the genetic perturbations are unknown.

For efficient organoid analysis, and to overcome the challenge of variable organoid size and variable lower bounds of organoids in the image stacks, we developed a novel algorithm to find entire organoids automatically in a large image stack (Fig 5D) and restricted our analysis to organoid-containing volumes for which the individual micrographs were well focused. To perform the 3D cropping of the entire multichannel organoid image stack, we used image slices from the nuclear (DRAQ5) channel. The 3D raw image stack was first projected on a 2D plane using focus stacking. Intensity thresholding and watershed transformation, located in X and Y the individual organoids in each focus stacked image. The Z-locations of the top and bottom planes containing useful data were identified by thresholding based on focus (Fig 5D). Focus based identification of planes makes intuitive sense as the focus of image planes degrades with distance from the lens for large objects due to light scattering and absorption. The images were then cropped in 3D using a 3D bounding box for each organoid. Phindr3D features were then calculated from all three fluorescence channels (Mitotracker, Lysotracker and DRAQ5) and together with the number of MV per organoid (panel F in Fig E in S1 Text) were used for clustering. This method served to reduce the image data by 200 fold (see Materials and Methods section for details).

Sammon mapping of Phindr3D features highlighted the heterogeneity of the resulting phenotypes (Fig 6B). Clustering control and oncogene-expressing MCF10A organoids identified seven phenotypes (Fig 6C and 6D) that cannot be distinguished using 2D features (Fig 6E) and were poorly differentiated using 3D segmentation and predefined features (Fig 6F). Moreover, 3D segmentation and feature calculation are very computationally intensive (Fig 6G). The spatial relationships and compositions of the clusters were visualized by a Sammon plot of the

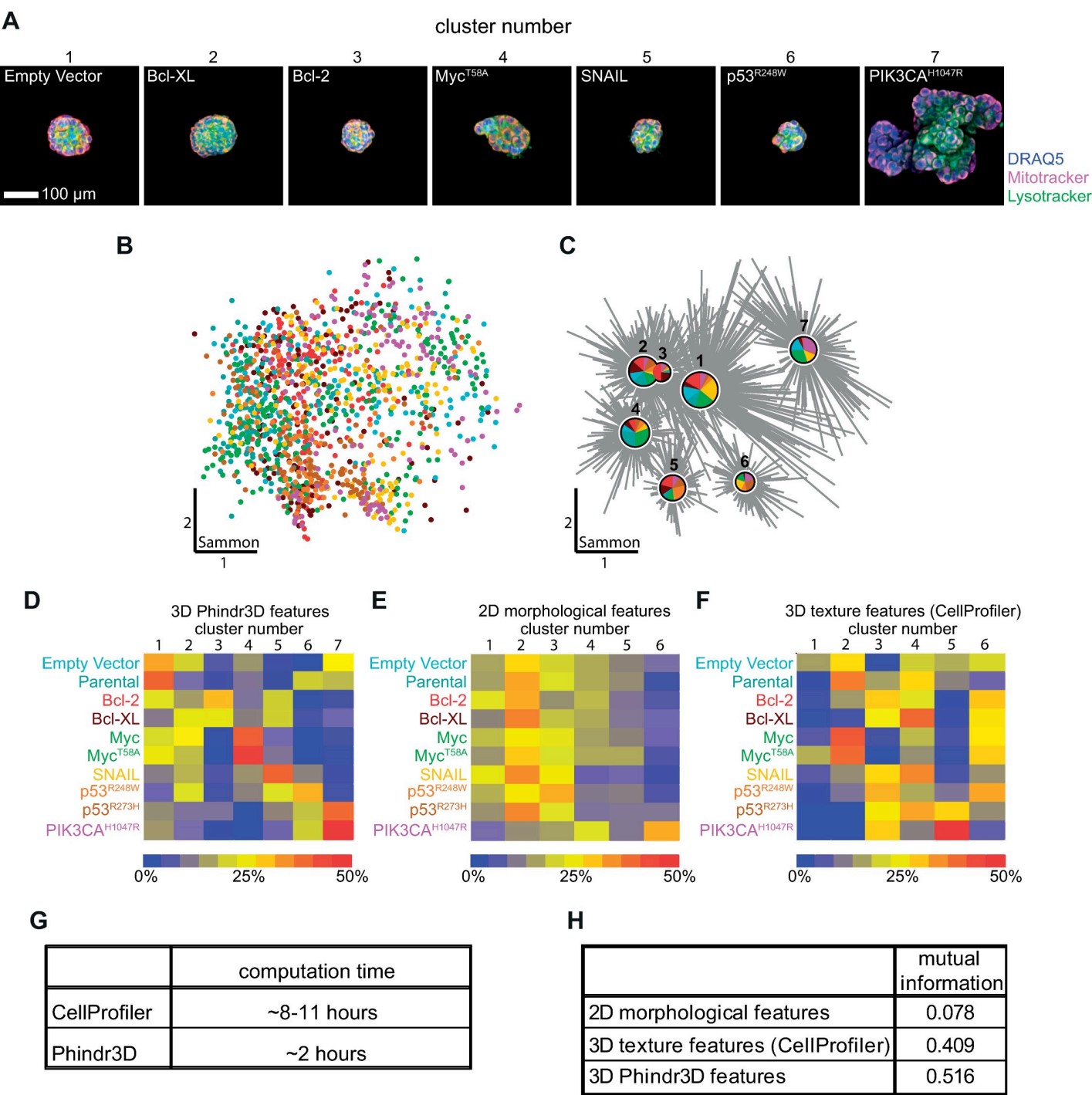

**Fig 6. Oncogene-driven morphological alterations in organoids. A)** Representative organoid images from each cluster with transgene and dyes indicated. **B)** 2-dimensional Sammon projection of feature space for all organoids, **C)** clustering result and **D)** heat map of proportion of transgene expressing MCF10A organoids in each cluster. **E)** Clustering after extraction of 2D morphological features results in low information separation of the data. Morphological features assessed included 2D Area, Major Axis Length, Minor Axis Length, Eccentricity, Equivalent Diameter, Solidity, Extent, Convex Area, and Perimeter from maximum-projected 2D regions. As above, affinity propagation was used to clustering the 2D morphological features. Phindr3D methods for determining meaningful cluster numbers by affinity propagation clustering were used as in panels a-c) and resulted in six distinct clusters for the 2D data. However, clusters 2 and 3 and to a lesser extent cluster 1 contained the majority of organoids while only PIK3CA^H1047R-expressing organoids formed a distinct cluster (cluster 6). **F)** After 508 3D texture features were calculated from the nuclear and organoid areas identified using the 3D segmentation algorithm in CellProfiler 3.0.0, the data were clustered by affinity propagation using the Phindr3D method for determining meaningful cluster numbers. Clustering again resulted in six distinct clusters, with clusters 2, 3, 4 and 6 containing large proportions of the organoids. As in D) PIK3CA^H1047R-expressing organoids formed a distinct cluster (cluster 5). Empty Vector-expressing organoids were present in all

clusters except cluster 3 in similar proportions. **G)** Computation time for 3D feature extraction was measured on a high performance laptop computer. Feature extraction alone required ~2 hours with Phindr3D and ~8 hours with CellProfiler. In addition, CellProfiler required reformatting the data files to TIFF stacks, which took ~3 hours. **H)** The utility of the extracted features for clustering to identify unique morphologies within the data was quantified using mutual information (MI). Low MI (i.e. values closer to 0) indicates that clustering using 2D morphological features results poorly predicted allocation of organoids while high MI indicates high probability that clustering using 3D features accurately reflects real differences in the data for images of oncogene-expressing organoids driving allocation to specific clusters. *For Sammon plot (B,C): colors indicate transgenes (from D-F), each data point represents one 3D image stack, cluster centers are shown as pie charts indicating the proportion of treatment points within each cluster. The diameter of the pie charts represents the number of data points in that cluster with lines linking data points to cluster.*

clustered data (Fig 6C). To account for differences in the numbers of organoids expressing each oncogene the compositions of the clusters are displayed as a heat map (Fig 6D). The number and percentages of organoids for each oncogene assigned to the different clusters are shown in Table A (S1 Text).

Importantly, projection based analysis of these images using standard 2D features, as routinely used to analyze 3D organoids [19] provided insufficient information to differentiate organoids expressing different oncogenes (Fig 6E and 6H). Moreover, for nuclear segmentation in 3D as recently implemented in various applications [7,20], the segmentation and feature extraction process was very time consuming (Fig 6G), and provided inferior results as defined by information content, compared to the Phindr3D cell segmentation-independent data-driven features of 3D microscopy images of organoids (Fig 6H). Deep learning analysis of maximum intensity protected images of the oncogene-expressing organoids using a convolutional neural network (CNN) provided results largely comparable to the Phindr3D analysis, albeit using 500 of the 1330 image stacks for training of the CNN (Fig F in S1 Text) compared to the 10 needed for Phindr3D analysis. Moreover, implementation of CNN analyses for these data was not straight-forward as there is currently no deep learning software with a user interface that supports data exploration for 3D microscopy data.

When using 2D features, the low quality allocation into clusters is evident (Fig 6E) as a large proportion of organoids for all oncogenes except PIK3CA$^{H1047R}$ were assigned to clusters 2 and 3. On the contrary, using Phindr3D features (Fig 6E), EV and Parental organoids had high proportions of organoids in cluster 1, while SNAIL had a high proportion of organoids in cluster 5 and Myc and Myc$^{T58A}$ had the highest proportion of organoids in cluster 4. None of these differences were identified using 2D features. Using CellProfiler 3D features resulted in clusters of intermediate specificity (Fig 6H). Only for PIK3CA$^{H1047R}$ were the phenotypes sufficiently divergent from Empty Vector and Parental that they clustered independent of the remaining organoids when using any set of 3D features. In summary, 3D analysis of this 3D data set using Phindr3D features provided superior inference than projection-based 2D features or 3D texture features based on cell segmentation (Fig 6H) as judged by allocation of oncogene-expressing organoids to specific clusters. Importantly, our data driven method also clustered the data equivalently to the results obtained using deep neural networks that required many more confocal stacks for training (Fig F in S1 Text). Moreover, using Phindr3D the analysis can be run on a standard desk-top computer in a computation time efficient manner (Fig 6G) rather than depending on a cloud-based solution.

As expected, due to the heterogeneity inherent in the MCF10A organoid system expression of the oncogenes resulted in organoids clustered using Phindr3D features with variable proportions of the images in each of the clusters. Nevertheless, several of the clusters were highly enriched for organoids expressing specific oncogenes. For example, cluster 7 contains most of the PIK3CA$^{H1047R}$-expressing organoids (51%). Cluster 7 also contained a large fraction of the organoids expressing p53 with the activating R273H mutation. These results are consistent with the relatively high information content for the clusters generated from the Phindr3D features by affinity propagation.

A challenge with the results of any clustering experiment is identifying what drives the formation of a cluster and how best to interpret the morphology represented by an individual cluster. Here the organoids in Cluster 7 had the highest 2D area and number of MV (an indicator of organoid volume, panel E,F in Fig E in S1 Text). Cluster 6 consisted of organoids with the highest intensity in Mitotracker staining (panel B in Fig E in S1 Text, Table B in S1 Text), possibly due to mitochondrial hyper-polarization. Cluster 4 had a higher median nuclear intensity compared to the rest (panel C in Fig E in S1 Text, Table B in S1 Text) and may be a result of uneven staining by DRAQ5 in Matrigel. Measurement of intensity and morphology features for organoids in the other clusters failed to identify possible driving features (Fig E in S1 Text). Typically, MCF10A organoids are assessed as 'normal' by an expert through manual identification of acinar structures from 2D morphology images. Thus, some relatively objective manual classification criteria exist that could be used to analyze the parental and EV only transfected controls. To determine whether affinity propagation clustering segregated normal acinar structures within the organoid population, we manually identified acinar structures from EV and parental organoids using the nuclear channel alone by examining bottom, middle and top z-planes for each organoid image stack and categorized the organoids into "normal" or "abnormal" based on expert evaluation of roundness and hollowness. For manual inspection of organoids, we only used organoids that had consistent DRAQ5 staining for all z-planes in the image stack and that were segmented correctly. We then determined that most organoids manually identified as 'normal' were assigned by affinity propagation to cluster 1 (Fig 5). Consistent with this result cluster 1 included the highest proportion of empty vector and parental organoids (Fig 6D, Table A in S1 Text). Taken together we conclude that cluster 1 contains "normal" organoids. While enriched in cluster 1, empty vector and parental organoids were also assigned to other clusters. These results are consistent with manual scoring suggesting that only 33% of the empty vector and 42% of the parental organoids exhibited a 'normal' morphology.

Apriori it was unknown whether expression of different oncogenes would result in different phenotypes in the organoids as this type of comparison has not been made previously. For this reason, we selected some of the oncogenes anticipating that they would result in similar phenotypes. For example, Bcl-2 and Bcl-XL are both anti-apoptotic proteins that might be expected to generate similar phenotypes. Although it is not obvious what features drive clustering for the other organoids it appears that the clustering does reflect the biology of the oncogenes expressed. Organoids expressing anti-apoptotic proteins (Bcl-2/Bcl-XL) were uniquely assigned to cluster 3 and to multiple other clusters (Fig 6D). Consistent with both Bcl-2 and Bcl-XL functioning to inhibit apoptosis their distribution across clusters was similar. Thus, expression of these anti-apoptotic proteins resulted in some organoids exhibiting characteristics not observed for either normal or more aggressively transforming oncogene expressing organoids. Moreover, the Myc-expressing and the stabilized mutant $Myc^{T58A}$–expressing organoids were also distributed similarly, as expected [21] since both proteins were expressed to high levels (panel A in Fig D in S1 Text). The highest proportion of SNAIL-expressing organoids was in cluster 5 (42%), consistent with SNAIL being known to cause epithelial-to-mesenchymal transition (EMT) in MCF10A cells [22]. The distribution of the two p53 mutants share some overlap (cluster 6) but are otherwise distributed in distinct clusters (Table A in S1 Text), which is consistent with different p53 mutants having distinct oncogenic potential [23]. Further analysis revealed close agreement of many of the above results with prior studies on oncogene function in malignant transformation of MCF10A mammary acinar organoids [24,25]. These data demonstrate that Phindr3D enables objective quantification of phenotypes despite the heterogeneity in MCF10A organoids.

## Discussion

Here, we have demonstrated that cell segmentation independent identification of data-driven features using a shallow learning method enables multivariate phenotypic analysis of large 3D microscopy data sets of complex (live) biological culture systems including neuronal cultures and organoids. Further, we implemented methods to overcome the challenges of identifying a biologically meaningful number of clusters to analyze a given data set, and for identifying individual organoid volumes in large image stacks.

We anticipate that Phindr3D will be of utility for easy quantitative phenotypic high content analysis of large hard-to or impossible-to segment (live cell) confocal 3D microscopy data sets such as neuronal cultures or organoids as used herein. It allows facile and rapid data exploration as well as quantification. Because Phindr3D discovers image features based on pixel intensity frequencies, it will not be able to distinguish different textures with the same distribution of normalized pixel intensities. While Phindr3D is not designed to distinguish cellular morphology or shape, some morphological information can be obtained by quantifying the number of MV in certain cases (panel F in Fig E in S1 Text). Furthermore, the results generated by Phindr3D are sensitive to the spacing between the z-slices of the 3D image stacks. For Phindr3D to perform best, z-slices should be spaced at Nyquist resolution. Larger spacing of the z-slices will lead to loss of resolution and thereby affect the output (see Results and Materials and Methods sections "Phindr3D image processing" for further details and prerequisites).

As an example of an analysis with an unknown answer that cannot be easily carried out using conventional approaches, dense cultures of primary cortical neurons were exposed to small molecule BH3-mimetic inhibitors of Bcl-XL (A1155463, A1331852), Bcl-2 (ABT-199) or the triple inhibitor of Bcl-XL, Bcl-2 and Bcl-w (ABT-263). As the effect of these inhibitors on dense cultures of primary cortical neurons is unknown, these experiments represent an entirely novel examination of the importance of different anti-apoptotic proteins to normal neuronal morphology (Figs 3 and 4). Furthermore, the ability to perform concentration response analyses using biochemical approaches is limited by the availability of primary neurons. However, previous image based analyses using uptake of nuclear dye in cultured hippocampal or cerebellar granule neurons have shown that neurons undergo apoptosis in response to inhibition of anti-apoptotic proteins with high concentrations of the early generation Bcl-2/Bcl-XL/Bcl-w inhibitor ABT-737 [26]. Furthermore, inhibition of anti-apoptotic Bcl-2 family proteins using high concentrations of ABT-737 but not the Bcl-2 specific inhibitor ABT-199 led to axonal degeneration in dorsal root ganglia explants [27]. These previous studies examining simultaneous inhibition of multiple anti-apoptotic proteins with ABT-737 or by expression of the BH3-protein PUMA provided background rationale to expect changes in neural morphology in the dense cultures of primary cortical neurons used here for at least exposure to ABT-263 as it is a close relative of ABT-737.

Our results clearly demonstrate that inhibition of Bcl-XL, Bcl-2, or Bcl-XL/Bcl-2/Bcl-w result in distinct neuronal phenotypes (Fig 4A). Taken together, using data-driven features and automated estimation of meaningful cluster numbers it was possible to quantify in dense cultures the responses of primary cortical neurons to treatment with small molecule inhibitors of anti-apoptotic proteins. These results suggest that using data driven features to analyze 3D images of dense neuronal cultures will allow quantitative analyses of a wide variety of small molecule effects and by extension characterization of genetic alterations. Moreover, the rapid and computationally facile data analyses provided by our Phindr3D method should enable functional genetic screening in primary cells.

As an example of complex functional genetics, we investigated the impact of oncogene expression on the phenotype of human mammary acinar organoid cultures derived from the non-transformed breast mammary gland epithelial MCF10A cell line. Using our Phindr3D method, we rapidly analyzed 1330 MCF10A mammary acinar organoids expressing different oncogenes. This analysis also served as a model for the analysis of morphogenesis in organoids and spheroids in general. Despite significant advances in 3D image analysis and substantial shortcomings in the analysis of 3D data using 2D features (Fig 6, Fig E in S1 Text), 2D projection-based measurements are still standard practice to analyze 3D assays [19,25,28,29]. Moreover, the lack of a facile and time efficient system for automated analysis has limited investigations to relatively small numbers of organoids, often less than 100 [25,29].

Among others, MCF10A mammary acinar organoids have been used to investigate the role of apoptosis signaling in tumorigenesis [24] or oncogene-mediated translocation of tumor cells [30]. However, unlike previous work, here we objectively quantified the phenotypic heterogeneity inherent in the cell line and that resulted from constitutive expression of oncogenes, including anti-apoptotic proteins. We found that based on the proportion of organoids in each cluster, both oncogene-expressing mammary acinar organoids and the EV and parental organoids displayed various morphologies. MCF10A cells undergo spontaneous oncogenic transformation the extent of which depends on passage number, precise lot of Matrigel and other factors. Since the organoids are clonally derived, it is therefore not surprising to find a relatively small proportion of EV or parental organoids in clusters predominantly containing oncogene-expressing organoids. Similarly, it was expected that organoids derived from MCF10A cells expressing exogenous oncogenes would be found in various clusters including with the empty vector controls due to variable levels of oncogene expression in individual organoids. That Myc- and Myc$^{T58A}$-expressing mammary acinar organoids showed a similar distribution across the clusters we identified in this data set suggests the two proteins are functioning similarly in this context. Previous comparisons of MCF10A acinar structures in response to ectopic MYC- and Myc$^{T58A}$ have reported a similar 3D transformation potential of these two MYC alleles when expressed at elevated levels [21,31] as seen here (panel A in Fig D in S1 Text). Only when ectopic levels are at low basal levels of MYC expression can the elevated transformation potential of Myc$^{T58A}$ be scored [25].

Despite having just a 'negative' control (parental and empty vector organoids, EV), using Phindr3D we were able to identify different 3D phenotypes of the mammary acinar organoids based on standard fluorescent stains without classification or manual supervision. Indeed, classification of this data is of limited value because both parental and EV organoids were highly heterogenous with only roughly a third conforming to predefined features for 'normal' acinar structures. Nevertheless, the proportion of organoids in each cluster was very similar for both the EV and the parental MCF10A organoids, demonstrating that cell segmentation-independent data-driven Phindr3D features can accurately capture the heterogeneous phenotypes inherent to this organoid model. Another advantage of Phindr3D features is that because they describe the 3D texture of an organoid, they can be easily combined with morphology features such as the number of voxels and/or conventional projection-based analyses such as the 2D area as used here for the MCF10A organoids. Thereby, Phindr3D feature extraction and unsupervised cluster analysis enabled rapid multivariate 3D analysis of 1330 organoids on a standard desktop computer.

Direct quantitative comparison using the mutual information in the clustering results demonstrated the importance of phenotypic profiling in 3D rather than using shape descriptors or predefined 3D features for the analysis of such heterogeneous image data (Fig 6H). Furthermore, the use of data driven features without segmentation of individual cells or nuclei outperformed the use of predefined features both in terms of information content in the resulting

clusters and the computational resources needed (Fig 6). Our method constitutes a major practical advance because the computed assignments represented the biology while preserving the heterogeneity of the underlying data. Indeed, tumor heterogeneity is a roadblock for targeted therapies in cancer [32] and here we provide a means for phenotypic profiling of complex organoid systems that more accurately captures the inherent heterogeneity than is possible with other current methods (Fig 6).

Applying the data driven cell segmentation free Phindr3D method to classification and clustering is equally facile. That the same data can be easily analyzed by both methods is a significant advantage when there is the possibility that a novel phenotype may be generated in the data. It is a common problem with supervised machine learning that phenotypes unlike any of the training controls are arbitrarily and unpredictably assigned to classes. Indeed, that the morphological changes of neuronal cultures upon treatment with BH3 mimetics are captured by clustering but not by classification is a direct result of the supervised machine learning model (random forests) since the classification boundaries are solely dependent on the set of training images. In this case, the phenotype(s) induced by ABT-199 or ABT-263 were different from the cell death induced by either ActD or STS and thus the classifier did not assign the new phenotype to the "ActD/STS" class. Although clustering better accommodates novel phenotypes, identifying the number of clusters is often subjective. The automated method we devised for selecting an appropriate preference value was based on the relatively smooth transitions between clusters due to the similarities in fluorescence micrographs of neurons and the substantial 'noise' due to biological heterogeneity. Our experiments demonstrate the utility of automated preference value tuning and affinity propagation clustering to separate the data into meaningful groups without requiring known phenotypes. Likewise, the variability in phenotypes and the resulting lack of certain ground truth in controls in our mammary gland acinar organoid data are another example of how classification of this type of data is extremely difficult. Nevertheless, affinity propagation clustering using automated preference value tuning revealed both differences and heterogeneity in phenotypes after treatment with inhibitors of anti-apoptotic proteins or expression of various oncogenes that would not have been discovered using supervised classification methods.

Deep learning methods for various tasks in microscopy image processing are now commonly used [12], including cell projection mapping [33], detection of cells in images [34], and image cytometry [35,36]. However, while we find that a convolutional neural network run on maximum intensity projected images of mammary acinar gland organoids expressing various oncogenes generates a result comparable to Phindr3D as determined by mutual information (Fig F in S1 Text), the need for training data is substantially higher for the CNN to perform well. Here, we used 500 of the 1330 organoid image stacks for training while Phindr3D required as little as 10 image stacks to generate results of comparable information content. Furthermore, deep learning methods may, at least in the field of image reconstruction, also come at (undesired) costs [37].

In summary, we have demonstrated the utility of data-driven phenotyping of hard-to-segment multichannel confocal 3D microscopy data by combining unsupervised feature selection with automated identification of high information images and optimization of cluster number. Integration of classification, clustering, and data visualization algorithms facilitates rapid data exploration. Additional potential applications include HCA of pluripotent stem cell-derived microtissues or complex spheroid/organoid systems [38]. Unlike the heavy computational resources required for deep learning, 3D segmentation, and feature calculation, the use of shallow learning enables rapid quantitative exploration of complex cellular systems and quantification of novel phenotypes with subcellular resolution from multidimensional confocal microscopy images on a standard PC.

## Materials and methods

### Ethics statement

All animal breeding and handling was performed in accordance with local regulations and after approval by the Animal Care Committee at Sunnybrook Research Institute, Toronto.

### Primary neuronal cell cultures and culture treatment

Primary brain cortical neuronal cultures were prepared from embryonic day 15 C57BL/6J mouse embryos as previously described [39]. Neurons from one mouse embryo were considered as one independent biological replicate ("N") and were seeded at a density of 6–8*10^5 cells / $cm^2$ in 384 well plates (Greiner μclear) after coating with poly-d-lysine (Cultrex). For excitotoxicity experiments, plates were also coated with collagen-g (Merck-Biochrom) unless otherwise stated. Briefly, after separation from hippocampus and subcortical structures, cortices were washed twice with ice-cold PBS, digested with 1x trypsin for 15 minutes at 37°C, washed twice with ice-cold PBS and then resuspended with a flame-treated glass pipette in N-Medium (DMEM, 10% v/v FBS, 2 mM L-glutamine, 10 mM Hepes, 45 μM glucose). The dissociated cortices were gently pelleted by centrifugation (200 x g for 5 minutes), N-media was removed, and neurons were resuspended and cultured in Neurobasal medium (Thermo-Fisher Scientific) supplemented with B27 (ThermoFisher Scientific), 0.5 mM L-glutamine and 25 μM glutamate at 37°C, 5% $CO_2$. The medium was partially replaced on day 5 in culture with Neurobasal medium supplemented with B27 and L-glutamine. For experiments using the Bcl-2/Bcl-XL inhibitors, ABT199, ABT263, A1155463 or A1331852 (all from Selleck Chemicals) or actinomycin D, staurosporine (both ThermoFisher Scientific) or DMSO as controls were added to the culture medium on day 8 and cultures were incubated at 37°C, 5% $CO_2$ for 24 hours and then stained and imaged (see below). For excitotoxicity experiments, neurons were treated with 25 μM glutamate in $BSS_0$ (116 mM NaCl, 5.4 mM KCl, 0.8 mM $MgSO_4$, 1 mM $NaH_2PO_4$, 26.2 mM $NaHCO_3$, 10 μM glycine, 1.8 mM $CaCl_2$, 10 mM HEPES pH 7.4) for 30 minutes (37°C, 5% $CO_2$) after washing the cultures with PBS as described [14]. As a control, 10 μM MK-801 (10,11-dihydro-5S-methyl-5H-dibenzo [a,d]cyclohepten-5,10-imine, (2Z)-2-butenedioate) and 10 μM CNQX (1,2,3,4-tetrahydro-7-nitro-2,3-dioxo-6-quinoxalinecarbonitrile; both from Cayman Chemical) were added to $BSS_0$ containing 25 μM glutamate. Medium was pooled and added to cultures after the incubation period. Live-cell stainings were performed 24 hours after these treatments.

### Culture of MCF10A cell organoids

MCF10A parental cells were a kind gift of Senthil Muthuswamy and were cultured as previously described [40].

**Cell line generation and verification.**   Selected oncogenes (Bcl-2, Bcl-XL, Myc, $Myc^{T58A}$, SNAIL, $p53^{R248W}$, $p53^{R273H}$, $PIK3CA^{H1047R}$) were first cloned into pENTR4-FLAG (w210-2) (Addgene plasmid #17423) and sequenced. Inserts were subsequently transferred into pLenti-CMV-Hygro-DEST (w117-1) (Addgene plasmid #17454) via recombination using Gateway LR Clonase II Enzyme Mix (ThermoFisher Scientific). Lentivirus from pLenti-CMV-Hygro-DEST after recombination with an empty pENTR4-FLAG (w210-2) plasmid was used to generate empty vector (EV) cells. MCF10A cells seeded at a density of $1x10^6$ cells per 10 $cm^2$ dish and grown for 24 hours. Cells were then transduced with 1 mL of virus, 4 ml of media and 30 μL of polybrene (0.8 mg/ml) overnight for 18 hours. Cells were then washed with PBS and 10 ml of fresh media was added for 24 hours. Following incubation, cells were selected with 200 μg/mL hygromycin. For Western blotting, cells were harvested in SDS lysis buffer and

loaded onto a 10% acrylamide gel. Immunoblots were probed using anti-FLAG (F1804, Sigma), anti-AKT (#9272, Cell Signaling), anti-p-AKT (S473) (#9271, Cell Signaling), anti-Actin (A2066, Sigma) and with secondary IRDye 800CW (LIC-926-32210, Licor) and Licor IRDye 800CW (LIC-926-68071, Licor) antibodies and visualized on the Licor Odyssey Classic.

**Matrigel plate preparation.** Frozen Matrigel was thawed on ice. 90 μl of cold serum-free DMEM/F12 medium (ThermoFisher Scientific) were put in each well of a 96 well plate (Corning #3615). Using ice cold pipette tips, 10 μl of cold Matrigel (Corning) were carefully layered underneath the medium so that the Matrigel formed a layer at the bottom. Edge wells were excluded from Matrigel addition. The plate was then incubated at 37° C for 1 hour and then 90 μl of medium was carefully removed without touching the Matrigel layer.

**Cell seeding.** 3D cultures of MCF10A were generated as previously published [40]. The cells were seeded on Matrigel-coated plates (see above) at 500 cells per well in 40 μl assay medium: DMEM/F12 supplemented with 2% horse serum (Invitrogen), 5 ng/ml EGF (Sigma-Aldrich), 0.5 μg/ml hydrocortisone (Sigma-Aldrich), 100 ng/ml cholera toxin (Sigma-Aldrich), 10 μg/ml bovine insulin (Sigma-Aldrich), 1% penicillin/streptomycin, 2% Matrigel (Corning Life Sciences). For each oncogene expressing cell line, 6 replicate wells were seeded per plate. The plate was stained and imaged after 12 days in culture with media changes every 3 days. The entire assay was repeated 3 times.

## Live cell staining

Neurons were stained with Alexa488-labeled annexin V, which was purified and labeled according to established protocols [41], 500 nM MitoTracker Red CM-H2XRos (ThermoFisher Scientific), and 1 μg/mL Hoechst 33342 (Cell Signaling Technologies, experiments with Bcl inhibitors) or 5 μM DRAQ5 (BioStatus, excitotoxicity experiments) as nuclear stain for 30 minutes at 37°C and 5% CO2. MCF10A organoids were stained with 5 μM DRAQ5, 500 nM Mitotracker Red and 100 nM Lysotracker for 30 minutes at 37° C and 5% CO2. High-content 3D microscopy was performed immediately after staining.

## High-content microscopy

Live cell high-content 3-dimensional microscopy of neurons (Bcl Inhibitors experiments) and MCF10A organoids was performed on an Opera Phenix (PerkinElmer) or on an Opera QEHS (PerkinElmer, for neuronal excitotoxicity experiments) spinning disc confocal microscope. On the Opera Phenix we used 375 nm (50 mW), 488 nm (150 mW), 535 nm (100 mW), and 635 nm (140 mW) solid state lasers, a 5x (NA 0.16) air objective, and 20x (NA 1.0) water and 63x (NA 1.15) water objectives and four sCMOS cameras (4.6 MP, 16-bit), controlled by Harmony v. 4.4 (PerkinElmer). On the Opera QEHS we used 488 nm, 561 nm and 640 nm solid state lasers, a 60x water immersion objective (NA = 1.2), three peltier-cooled high-resolution CCD cameras (1.3 MP, 12-bit), controlled by Acapella 2.0 (PerkinElmer). Image acquisition was performed unbinned at 37° C and 5% $CO_2$.

For the experiments using the inhibitors of apoptosis inhibitory proteins Bcl-2 and Bcl-XL, 18 z-slices were obtained per image stack with the 63x water objective, with a distance of 0.7 μm between each z-slice. For the excitotoxicity experiments, 21 z-slices were obtained per image stack with a distance of 0.7 μm between each z-slice.

For organoid imaging, a low resolution (5X) scan was performed ("PreciScan") to identify the 2D (X and Y) location of the organoid using the nuclear channel. The location was then used for 3D imaging with the 20x water objective. The distance between each z-slice was set to 5 μm. The number of slices varied per plate due to variations in the thickness of the Matrigel and was determined manually based on the nuclear channel alone. Similarly, manual

adjustments were made in the start height for some of the wells to enable recording of most of the organoids. Images of organoids from all plates were pooled together for further processing.

## Image processing and analysis

For figures, maximum intensity projection (MIP) images from representative image stacks prepared in Harmony 4.4 are shown for illustrative purposes only and were not used for analysis. For presentation of MCF10A organoid gallery images, MIP images were placed on equally sized black canvases.

**Neuron image pre-processing.**   Z-stacks from the Bcl-2/Bcl-XL inhibitor experiments were adjusted manually to contain equally spaced z-volumes due to variable heights of the first slice that contained in-focus information, resulting in 9 slice-containing z-stacks throughout. Images from the Opera QEHS were converted from FLEX to 16-bit TIF format in Acapella 2.0 (PerkinElmer) without any further modification and subsequently processed as described below.

**Organoid image pre-processing.**   Since different organoids in the same image field were located at different z planes, it was necessary to identify individual organoids in 3D. Organoid contour segmentation and downstream computation of start and end Z planes for the organoids was accomplished using the nuclear channel and the following image processing steps:

1. *Focus Stacking*: The first step was to identify the 2D location of each organoid. Individual slices of the 3D image were filtered using median filtering (kernel size 3x3). For each slice, we computed the standard deviation of the pixel intensity within an 11 by 11 pixel window. Then the image gradient magnitude was obtained for each slice in the convoluted image stack. To calculate the projected image, the z plane with the maximum gradient magnitude was computed for each pixel. Higher gradient magnitude implies significant changes in the pixel neighborhood. Such variations can only be observed when the specific pixel is in focus. We then used the intensity of the pixels (from the original image before gradient transformation) with the highest local gradients and projected them onto a new 2D image. Both the focus stacked intensity image and the best focus planes for each pixel were stored for later use.

2. *Organoid contour segmentation*: From the focus stacked intensity image, individual organoid contours were segmented using a series of steps. Briefly, the image was filtered using a disk with radius 75 pixels, after which a threshold was computed using the maximum correlation thresholding (MCT) algorithm [42] to separate foreground from background pixels and any holes in the binary image were filled. Next, the organoid region was identified and segmented using a watershed algorithm. Finally, the x-y bounding box was computed for each organoid.

3. *Identify top and bottom planes for each organoid*: We applied the contour segmentation mask to the best focus plane image obtained from step 1. Each pixel in this image contains values ranging from 1 to the maximum number of z planes. Within the segmented region for each organoid, we computed the density curve from normalized histogram pixel values from the best focus image where each value corresponds to individual image planes of the original image stack. The histogram is a measure of the contribution of each in focus plane for the organoid. If there are n planes, the random chance of any plane being in focus and thus contributing significantly to the histogram is 1/n. We used this value to find the lower and upper plane. We further increased the number of planes by 2 on each side and a new organoid image stack was saved for analysis. See Fig 5D for details.

We saved cropped image stacks using a 3D bounding box based on the x-y bounding box from step 2 and lower and upper z-planes from step 3. Thereby, we decreased the size of this data set from 3 TB to 15 GB. Before computing Phindr3D features we manually removed all the organoids that were low in DRAQ5 intensity. We used Phindr3D features (MV categories) and in addition, the number of MVs in each organoid as another feature to cluster these data. The organoid contour segmentation workflow is provided as a stand-alone program separate from Phindr3D on GitHub (https://github.com/DWALab/Phindr3D).

## Phindr3D image processing

Phindr3D is available on GitHub (https://github.com/DWALab/Phindr3D). In the Phindr3D method unsupervised clustering is applied iteratively and pixels are combined at different hierarchical levels to calculate image features without the need for explicit cell segmentation and object identification. The prerequisite for Phindr3D analysis is that microscopy is performed close to Nyquist resolution, and that all z-stacks (across all replicates) have the same characteristics (number of top/bottom slices with out-of-focus or no information). Given a multichannel 3D microscopy data set, these computations are first performed on a set of randomly selected or user defined training images (a minimum of 5 images per condition) to get image parameters, which are then applied to the entire dataset to compute image features. The number of training images is set to 10 by default. For a given 3D image dataset, the following steps compute image features.

**Rescaling pixel intensities.** To avoid intensity bias from any channel in downstream processing, intensities for each channel are scaled between minimum (chmin) and maximum (chmax) values. Typically, the bias would occur due to differences in the intensity range among the different channels acquired. This difference in the intensity range would drive the downstream unsupervised clustering and bias the pixel categories towards a particular fluorescent marker/channel. The rescaling values are determined by taking the lower and higher 5% quantile values for each image and each channel from the training set. The final minimum and maximum is determined as the median of minimum and maximum intensity values, respectively. The chmin and chmax values are then applied to all images including the images from the training set for rescaling:

$$I_{rk} = \frac{I_{ok} - chmin_k}{chmax_k - chmin_k} \qquad (1)$$

Where $I_{rk}$ is the rescaled image for the kth channel, $chmin_k$ and $chmax_k$ are the respective minimum and maximum intensity values for kth channel image and $I_{ok}$ is the original image. Since the same scaling values are applied across the image dataset, the original intensity differences between the images of different treatments are preserved. The user can also calculate scaling for any columns of the metadata separately. For example, to analyze multiple replicate plates, it would be necessary to remove or reduce the effects of differences in image intensities arising from different plates. In such a case, the user can set the scaling values for each plate separately.

**Image thresholding.** Individual image channels are thresholded to distinguish foreground and background pixels using the MCT algorithm [42]. Briefly, pixel-wise intensity correlation is computed between a series of thresholded images and the original image. The best threshold is the one with the highest correlation. To avoid calculating a threshold for every image slice of individual channels in the 3d stack, a common threshold parameter is determined from the training set. For every image from every channel in the training set, a threshold for individual slices with the 3d stack is computed. This gives a set of threshold values. The

final threshold for every channel is determined as the median of the threshold values. The user is given a choice to modify this value from lower quartile to upper quartile. Pixels less than the threshold values are considered to be background pixels.

**Determining pixel categories.** This step is used to assign a category for every multi channel pixel (i.e. same pixel location across all the channels). Thus, every pixel can be thought to be a multidimensional vector, where the number of dimensions equals the number of channels. This step is done for every z plane in the 3D stack. A pixel is considered a background pixel if all the channels in the same x-y location are below the threshold. To determine pixel categories, only foreground multidimensional pixels from the training images are clustered to identify pixel categories. Scaling reduces bias in clustering by decreasing the influence of any particular channel on the clustering results, since large intensity values can dominate the outcome of clustering. Pixel categories are computed using k-means clustering where the number of clusters $k_p$ is user defined. An empirical default value of 15 is used for k. Each pixel category represents a specific combination of intensities from the different channels. These pixel centroids are later used to assign pixels from every query image to the nearest centroid based on Euclidean distance. The original 3D multichannel image stack is now transformed into a 3D pixel category image stack $I^P$.

**Determining supervoxel categories.** A supervoxel (SV) is defined as a distinct block of neighboring pixels in 3D in the pixel category image stack ($I^P$) obtained from clustering pixels in the previous step. For any supervoxel, the number of background pixels are computed. Supervoxels with more than 50% background pixel category are considered as background supervoxels and are not used for analysis. The threshold value for background supervoxels and the size of the supervoxels can be user-defined. Default values are 10, 10, and 3 pixels in the x, y, and z directions, which we found to work reliably with all of the different datasets we used (panel F in Fig C in S1 Text). Every SV is characterized by the relative frequency of pixel categories within the SV. Thus, a supervoxel is a vector of length k ($SV_k$):

$$SV_k = NP_k / \sum_i^{k_p} NP_i \tag{2}$$

Where $NP_k$ is the number of pixels of category k, and $k_p$ is the total number of foreground pixel categories. SV centroids are then computed using k-means clustering using foreground supervoxels with a user-defined number of categories ($k_{SV}$) using training images. SV centroids are then used to assign all foreground supervoxels to categories using the nearest neighbor rule based on the Euclidean distance to the SV centroids. The pixel category image stack is now transformed into a SV category image stack. The dimensions of the SV category image stack are scaled by $m_x$, $m_y$ and $m_z$ in the x, y and z dimension, where $m_x$, $m_y$ and $m_z$ are the size of the supervoxel in pixels in x, y and z directions. For example, a 1000 by 1000 by 3 image is transformed into 100 by 100 by 1 using a supervoxel of size 10 by 10 by 3.

**Determining megavoxel categories.** A megavoxel is defined as a block of neighboring supervoxels in the x, y and z dimension from the supervoxel image stack. The size of a megavoxel is user-defined with default values set to 6, 6, and 2 in x, y, and z dimensions. The analysis is robust to the choice of supervoxel and megavoxel sizes (panel F in Fig C in S1 Text). A megavoxel can be considered to be a vector ($MV_k$) of $k_{SV}$ dimensions, where $k_{SV}$ is the number of foreground supervoxel categories:

$$MV_k = NSV_k / \sum_i^{k_{SV}} NSV_i \tag{3}$$

Where $NSV_k$ is the number of supervoxels from category k and $k_{SV}$ denotes the total number of supervoxel categories, and $NSV_i$ denotes the number of supervoxels of category i within the megavoxel.

Megavoxels containing at least 50% foreground supervoxels are considered foreground megavoxels. MV categories are computed using k-means clustering with user value for number of foreground megavoxel categories $k_m$. Similar to the supervoxels, the MV centroids are then used to assign MV categories to any query megavoxel using the nearest-centroid rule.

**Computing image stack features.** Each multi-channel image stack is now described by the relative frequency of foreground MV categories. These categories are defined as the image stack features. Thus, the number of image features is equal to the number of MV categories.

$$I_k = NMV_k / \sum_i^{k_m} NMV_i \tag{4}$$

Where $I_k$ is the image, $k_m$ is the number of foreground MV categories and $NMV_i$ denotes the number of mega voxels assigned to category i. Note that by omitting the number of background megavoxels to compute image features, blank image regions do not affect the image features.

**Visualization of feature space.** Visualization of the extracted features is performed using one of the several algorithms [6,43] from van der Maaten's toolbox (https://lvdmaaten.github.io/drtoolbox/, last accessed on September 10, 2019), which is integrated into the Phindr3D code.

**Image classification.** Once image stack features are computed, image stacks can be classified using a random forest algorithm [44] that is trained on user-selected controls. The number of trees is fixed to 100 while the number of variables at each node of the tree is optimized internally using nested cross validation. The classifier output can be viewed within the software as well as exported to a tab delimited file for further statistical analysis.

**Image clustering.** For unsupervised clustering, Phindr3D uses the affinity propagation (AP) algorithm [17]. Briefly, this algorithm identifies exemplars within the data set, which intuitively denote different phenotypes. The number of exemplars is tuned using a "preference" value for each point, which refers to the ability of the data point to become an exemplar. The algorithm iteratively uses a voting method to determine the number of exemplars based on preference values. In the "auto" mode in Phindr3D, affinity propagation is run over a range of preference values obtained using the lower and upper bound of preference values, and the number of exemplars against the preference values is plotted. In Phindr3D, we empirically identify the best preference value as the preference value after which the number of clusters grows exponentially with small changes in preference values using an angle-based method [16]. Briefly, a difference function is computed based on number of clusters and preference values given by

$$dF(m) = N(m-1) + N(m+1) - 2N(m) \tag{5}$$

Where N (m) is the number of clusters at the m$^{th}$ preference value. Since the number of clusters increase with increasing preference values, the first n maximal difference is preserved. Here we set n as 5. After this the point that makes the largest angle in the curve is calculated using

$$angle = \operatorname{atan}\left(\frac{1}{|N(m) - N(m-1)|}\right) + \operatorname{atan}\left(\frac{1}{|N(m+1) - N(m)|}\right) \tag{6}$$

The maximum angle gives the point of inflexion. We then use the point before as the number of clusters appropriate for the dataset. The graphical result of this calculation can be output and saved. This method was validated by determining affinity propagation performance on several reference cluster datasets (see next section). The algorithm can also be executed for a

user-defined number of clusters (e.g. if there is an expected number of clusters or if the calculated result fails due to noise).

## Phindr3D analysis of neuronal phenotypes

For analysis, image data from three plate replicates were pooled and Phindr3D features were calculated from the pooled images by setting the number of output features (i.e. MV categories) to 40. To correct for differences in image intensities on different imaging days, Phindr3D computed the required intensity scaling parameters for every channel for each replicate plate individually. Both a 2D principal component analysis (PCA) visualization of the 40-dimensional Phindr3D feature space for individual image stacks and for the averaged treatment centroids, and for each image stack for individual plate replicates are shown in Figs 3 and 4.

## Phindr3D algorithm validation

For validation of empirical estimation of number of clusters, we used multiple datasets both synthetic as well as published (synthetic 32 dimensional, IRIS, synthetic R15, bone marrow) datasets (see Fig B in S1 Text for details). For the synthetic datasets, we generated values for mean and variance that were randomly chosen from a uniform distribution. A Gaussian mixture model was then generated for the desired number of dimensions. Both of the above steps were repeated 30 times. After identifying the number of clusters automatically, we computed the adjusted RAND index (ARI) [45] for each dataset which indicates the agreement between the ground truth and the clustered data. The higher value of the ARI indicates better agreement. For each dataset we also computed the ARI for cluster assignments as if we had used the median of similarity values as preference as suggested by the authors of AP [17]. For the bone marrow dataset we used spearman correlation as similarity metric for input to AP, for all other datasets we used negative Euclidean distance as similarity metric for AP.

*Image artifact evaluation*: To test the effect of Gaussian noise on Phindr3D features, we added 2D Gaussian noise with different standard deviation for each image slice and each channel from neurons under control conditions ("buffer, BSS) and neurons treated with 25 μM glutamate treatment in BSS with increasing standard deviation of the Gaussian kernel. A 10-fold cross validation was done using random forest classifier and the Cohen's kappa was measured for each run. This was then plotted in panel B and C in Fig C in S1 Text. In a similar process Gaussian blur with different standard deviations of the Gaussian kernel was added and the 10-fold cross validated Cohen's kappa was plotted in panel D, E in Fig C in S1 Text.

## Morphological features for organoids

The 2D morphological features were obtained from the segmented 2D regions (mask) identified above. For each segmented region we calculated the following features:

a. 2D Area: Sum of all the pixels occupied by the 2D mask

b. Major Axis Length: Length of the major axis of the ellipse that has the same normalized second central moment as the mask

c. Minor Axis Length: Length of the minor axis of the ellipse that has the same normalized second central moment as the mask

d. Eccentricity: Eccentricity of the ellipse that has the same normalized second central moment as the mask.

e. Equivalent Diameter: Diameter of the circle that has the same area as the mask

f. Solidity: Ratio of area to convex area

g. Extent: Ratio of area to area of the bounding box for the mask

h. Convex Area: Sum of all the pixels in the convex hull of the 2D mask

i. Perimeter: Number of pixels along the border of the organoid

In addition to the above, we also computed "hollowness" for each organoid. To calculate hollowness for an organoid, we split the 2D binary mask for each organoid into two regions, the outer region (a ring region that was 40 pixels wide) and a central region (region of the mask excluding the ring region). For each organoid, we computed the average pixel intensity of the mid slice in the central region and in the outer region. For clustering using 2D morphological features only, we included features (a) to (i) as described above.

## CellProfiler analysis of organoids

For comparison with the Phindr3D method, we used CellProfiler 3.0.0 [7] for analysis of 3D organoids. We first reformatted the single plane TIF files into TIF stacks for each field and channel. For comparison, we analyzed the same set of organoids that were analyzed by Phindr3D and after compressing to 2D by maximum projection. For each organoid nuclei were identified in the TIF stacks, using the DRAQ5 image. Then whole organoid masks were obtained by morphological dilation of the nucleus region. The segmentation algorithm in CellProfiler resulted in oversegmentation of some of, the organoids. Therefore, over-segmented organoids were combined by computing the average values of the features weighted by the number of nuclei in each of the organoids after computing 3D texture features and the number of nuclei in each organoid. Finally, organoids with less than 10 nuclei (that possibly arose due to incorrect segmentation) were removed from the analysis. Overall, 508 features were extracted and normalized by z-score prior to clustering using the clustering tool implemented in Phindr3D. The same clustering tool was used for all analyses to take advantage of the automated determination of a meaningful number of clusters and to permit direct comparison of the utility of preconceived 3D features with the learned pixel based features in Phindr3D. Computing time was measured on an Alienware R15 high performance laptop computer (3.5 GHz Intel i7 CPU, 16 GB RAM, Windows 10) and compared to computation time of Phindr3D on the same machine.

## Deep learning analysis of organoids

Each 3D image stack was flattened to a 2D image using maximum intensity projection (MIP). Then, individual 2D oncogene images were cropped or padded to form a 256x256 pixel image. We created a 4 layer convolutional neural network (CNN) with 1 stride and rectifier linear unit activation. Max pooling was done after a convolutional layer. A flattening layer was added at the end along with a dropout layer. Finally, dense layers were added for final classification. 50 images from each individual class (total of 10 classes) of oncogenes were chosen for training (total 500 images) and 30 images per class were chosen for validation. We used Google Colab and Python Keras implementation for running deep learning. For data clustering, embeddings were computed by passing every image through the network and taking values from the dense layers of the trained network at the end. Nuclear intensities were computed separately for each 2D MIP image using the Hoechst channel. Intensities lower than 850 were removed from analysis. Finally, the embeddings were used to cluster data using affinity propagation. The number of clusters were chosen to match the clusters obtained from the Phindr3D analysis. The overall run time on the compute resources allocated by Google Colab was approximately 2 hours including training and computing the embeddings but excluding the required data preprocessing.

## Mutual Information between clusters and oncogenes

The mutual information between the oncogenes and clustering was computed as follows

$$MI(oncogene; clusters) = \sum_{i=1}^{m}\sum_{j=1}^{n}p(o_i, c_j)log\left(\frac{p(o_i, c_j)}{p(o_i)p(c_j)}\right) \tag{7}$$

Where MI is the mutual information, m is the number of oncogenes (m = 10), n is the number of clusters,

$$p\left(o_i, c_j\right) = \frac{Number\ of\ organoids\ of\ oncogene\ i\ in\ cluster\ j}{\sum_{i,j}number\ of\ organoids\ of\ oncogene\ i\ in\ cluster\ j} \tag{8}$$

The marginal probabilities are given by

$$p(o_i) = \sum_{j=1}^{n}p(o_i, c_j) \tag{9}$$

and

$$p(c_j) = \sum_{i=1}^{m}p(o_i, c_j) \tag{10}$$

## Reference 3D data set

The 3D autophagy data set and the curve fitting data from the original analysis [13] shown in Fig 2 were kindly provided by Alexander Schreiner, Karin Boettcher, Stefan Letzsch (all PerkinElmer Cellular Technologies, Hamburg, Germany), and Alex Kalyuzhny and Jodi Hagen (R&D Systems, Minneapolis, USA). Contact details are available upon request. The use of these data in this manuscript was approved by PerkinElmer Cellular Technologies and R&D Systems prior to manuscript submission. Briefly, HeLa, HCT116, and PANC cells were treated with increasing concentrations of chloroquine ranging from 1.6 to 100 μM to inhibit autophagic flux. The cells were subsequently fixed and stained for nuclei (DAPI) and cytosol (fluoro nissl green), and immunostained for p62/SQSTM using an anti-human SQSTM-1 antibody (R&D Systems, MAB8028) and a Northern Lights 557-coupled secondary antibody (R&D Systems, NL007). 3D automated confocal spinning disk microscopy was performed on a PerkinElmer Opera Phenix with a 40x (NA 1.1) water immersion objective [13]. For analysis using Phindr3D, each image was split into 16 distinct equally sized images to get an adequate number of training images for Phindr3D as well as random forests and the metadata was changed accordingly to increase the number of data points for classification. These new images were then used as input to Phindr3D. Concentration response curves were fit and $EC_{50}$s were calculated after classification with the positive controls (100 μM chloroquine) and a 4-parameter non-linear regression fit in GraphPad Prism 5.0.

## Statistics

Oneway ANOVA with Tukey-Kramer post-hoc test was calculated in Matlab 2017b for the data in Fig E in S1 Text. $\alpha < 0.05$ was considered as statistical significant difference. See individual methods sections above for details on further statistics used for analysis.

## Supporting information

**S1 Text. This supplementary file contains additional figures and tables.**
**Fig A. Small molecule inhibition of Bcl-2 family proteins in neurons. A)** Sample images of primary cortical neurons treated with the indicated compounds illustrate the difficulty in observational assessment of drug responses in dense cultures. Prior to imaging the cells were

stained with mitochondrial outer membrane potential-sensitive dye Mitotracker (red), the apoptosis indicator annexin V labeled with FITC (green) and the nuclear stain Hoechst (blue). To assess the cell death responses of cells visually it is important to be able to relate nuclear condensation to changes in mitochondrial transmembrane potential or annexin V staining for the same cell, which is not possible in images of neurons at this density. Alternative measures such as Western blotting are not amenable to high-throughput or the small number of cells in one well of a 384 well plate. Images are for visual inspection with channels scaled to equal intensities and were not used for analysis. *Scale bar*: 100 μm. **B, C)** Variation in phenotypic response of primary cortical neurons to different compound treatments from three plate replicates. Two dimensional principal component projection of the Phindr3D feature space of 3D multichannel images for the common treatments from the three different plates. Each point represents one 3D multichannel image. Here, we only analyzed conditions that were common across all plate replicates. These data allow visual confirmation of the minimal variability across three plate replicates, encompassing a total of nine biological replicates (i.e. cortical neurons derived from 9 separate animals). We further computed average values of Phindr3D features for each combination of compound treatment and concentration in each plate and computed Euclidean distances between the average profiles resulting in a distance vector for a plate. A high correlation implies treatments are distributed similarly for each plate. Next, we calculated the correlation of distance vectors between each plate. This resulted in correlation coefficient values of 0.74 (Plate1-Plate2), 0.85 (Plate 1 –Plate 3) and 0.88 (Plate 2- Plate 3), confirming minimal variability across the plate replicates. **B)** The colors correspond to the different treatment groups indicated at the right. **C)** The colors correspond to replicate plates as indicated at the right. **D) Classification based concentration response accuracy depends on training sets.** Concentration response of primary cortical neurons treated with different anti-apoptotic protein inhibitory compounds calculated based on classification with a random forests classifier trained on DMSO negative controls and 10 nM actinomycin D (ActD) and 10 nM staurosporine (STS) positive controls (10 nM ActD/STS). Experimental means are indicated with horizontal lines, dots represent individual 3D image stacks. Phindr3D features were extracted from pooled images from three replicate plates. Images were then classified as either DMSO (alive) or ActD/STS (dead). While increasing concentrations of A1155463 and A1331852 resulted in a corresponding increase in the number of images classified as ActD/STS, this was the case for increasing concentrations of ABT-263 only to a minor extent. The results for ABT-263 are most similar to those for untreated neurons (Medium). However, Phindr3D correctly identified inhibition of Bcl-2 by ABT-199 as a separate phenotype. Finally, the Phindr3D analysis trained on DMSO detected that some of the untreated cells (Medium) were not identical to the DMSO class illustrating the inherent heterogeneity within the cultures detected also by clustering (Figs 3 and 4) and the sensitivity of the method as well as the unpredictable classification of data that do not match a training class precisely. The discrepancy of this result compared to the high degree of cell death seen in the images (panel A) of the high concentrations of ABT-263 suggests that while visually recognizable as dead, the morphologies of these cells are not similar to cells treated with ActD/STS as further discussed in the main text.

**Fig B. Validation for the technique for determination of number of clusters in Phindr3D.** In all clustering algorithms, determining a meaningful number of clusters to represent the data appropriately is problematic and generally determined by trial and error. In AP clustering the number of clusters is determined by a variable called the preference value. For AP clustering, Phindr3D automatically estimates the number of clusters from the data by estimating the preference value beyond which the number of clusters increases exponentially (see Fig 5 and methods for additional details). This approach employs an angle based method for knee point

detection [1]. Empirical validation of the algorithm for determining a meaningful number of clusters was performed using published and our own gold standard data sets with established ground truths. The graphs show the adjusted RAND index (ARI), a measure for the accuracy of cluster estimation [2]. A value of 0 would indicate a random result and a value of 1 perfect agreement with the ground truth. Generally, for the type of data analyzed here, ARI values greater than 0.5 indicate successful clustering. **A, B)** ARI (left panel) and estimated number of clusters (right panel) for randomly generated Gaussian datasets with 10, 20 or 30 dimensions. For each dimension value, 30 different Gaussian datasets with 3 and 5 clusters were randomly generated using different means and variance each time. **A)** The number of clusters was estimated using different preference values for affinity propagation (AP) based on our algorithm for automated detection of the optimum preference value. The agreement between the true clustering and the estimated clustering was then expressed as the ARI (left panel) for every dataset. The right panel shows the resulting number of clusters when using our algorithm for automated detection of the preference value (n = 30 ±STD for mean ARI and mean number of estimated clusters). **B)** For comparison, we performed AP clustering on the same datasets using the recommended preference value pMedian [3], which is the median of similarity values between the data points, which results in ARI values (left panel) less than 0.5 in most cases. The right panel shows the resulting number of estimated clusters when using pMedian instead of our algorithm for automated detection of the optimum preference value (n = 30 ±STD for mean ARI and mean number of estimated clusters). **C)** Published datasets with different dimensions were clustered using the automatic clustering approach implemented in Phindr3D (black bars, estimated) or using pMedian (grey bars, pMed). After clustering, the ARI was computed in each case. While automated cluster estimation performed well for the real world data sets (IRIS, bone marrow), using pMedian instead performed better on the well separated synthetic data sets. Synthetic–a synthetic 32 dimensional dataset with 16 Gaussian clusters [4], IRIS–taxonomic iris flower data set with 4 dimensions and 3 clusters [5], R15 –a synthetic two-dimensional data with 15 clusters [6], Bone Marrow–microarray gene expression data of 999 genes (i.e. dimensions) from 38 leukemia patients [7] (i.e. clusters), see Materials and Methods for detail. In summary, these data demonstrate that the automated clustering approach in Phindr3D yields reliable results in a variety of different use cases.

**Fig C. Robust performance of data-driven Phindr3D image features. A)** Map of the phenotypic feature space of primary cortical neurons in response to excitotoxic glutamate treatment. Each dot represents one 3D image stack. This control data set was generated to investigate the robustness of the Phindr3D algorithm using an independent complex 3D data set of dense cultured neurons undergoing a well-defined response (excitotoxicity). Plotted is the two-dimensional t-SNE projection of Phindr3D features extracted from 720 three channel 18 stack image data of primary cortical neurons 24 hours after glutamate treatment. As controls, neurons were incubated in BSS0 (buffer) alone or the NMDA receptor antagonists (NMDA-Ra) MK-801 and CNQX were added with glutamate to ameliorate its effects. Prior to imaging, the cells were stained with Mitotracker, annexin V and DRAQ5. The data show a high degree of variability for the control condition (buffer) with (expected) overlap with the untreated condition and (undesired) overlap in the t-SNE representation into the treatment (glutamate) condition. As expected, the NMDA-Ra-treated condition occupies a different space in the t-SNE than both untreated and glutamate-treated neurons but with some overlap of the space occupied by the untreated and buffer conditions. This is consistent with a scenario where inhibition of glutamate receptors induces a phenotypic change in the cells that is different from the damage induced by glutamate treatment or the phenotypic alterations of the buffer condition. Despite the heterogeneity in the data, the image dataset was useful for measuring the effect of Gaussian noise, 3D Gaussian blur and different values for super-voxel and mega-voxel window size and

categories on the performance of Phindr3D features as shown in subsequent figure panels. *Untreated–neurons in medium; buffer–BSS0; glutamate– 25 μM glutamate in BSS0; glutamate + NMDA-Ra—25 μM glutamate in BSS0 +10 μM MK-801 +10 μM CNQX.* **B, C)** Effect of Gaussian random noise on Phindr3D classification performance to distinguish between neurons treated with 25 μM glutamate or incubated in BSS0 (Buffer). Gaussian noise added as random samples with a zero mean and increasing standard deviations resulted in degraded performance, i.e. the level of classification agreement assessed as Cohen's kappa between *No Noise* and *added noise* decreased. Cohen's kappa is a measure of the prediction performance of a classifier (a value of <0 indicates random assignment to a class; a value close to 0.7/0.8 indicates very good agreement). **B)** Sample single plane images of primary cortical neurons stained with Mitotracker (red), annexin V (green) and DRAQ5 (cyan) with increasing Gaussian noise from top to bottom. **C)** Separation of the 25 μM glutamate and BSS0 (buffer) classes assessed as Cohen's kappa over 10-fold cross-validation under the different random noise conditions. With no added noise, Cohen's kappa for these data was ~0.65 while the 25th and 75th percentiles were ~0.4 and ~0.85 reflective of the heterogeneity in the original data set. Adding random Gaussian noise decreased the spread in the data, presumably by reducing the influence of heterogeneous low intensity high frequency (small features) on the automated identification of image features by Phindr3D (see also panels D,E). As the standard deviation of the added noise was increased above 20, feature quality degraded and the separation of the 25 μM glutamate from the BSS0 (buffer) classes diminished. *Box plots show median (red) and 25th and 75th percentiles within the box, 10th and 90th percentiles as whiskers and outliers as individual data points, n = 10; scale bar 10 μm.* **D, E)** Effect of Gaussian blur on classification performance to distinguish between neurons treated with 25 μM glutamate or incubated in BSS0 (Buffer). Image blurring introduced by smoothing individual images using a Gaussian kernel with dimensions 21, 21 by 5 pixels and different standard deviations. **D)** Sample single plane images of primary cortical neurons stained with Mitotracker (red), annexin V (green) and DRAQ5 (cyan) with increasing added blur from top to bottom. **E)** Cohen's kappa over 10-fold cross-validation under the different blur conditions demonstrated that at standard deviation of 10 image blurring improved classification. This suggests that for this dataset the high frequency information (i.e. small structures) in the images is too heterogeneous to generate useful features (i.e. it is dominated by noise). *Box plots show median (red) and 25th and 75th percentiles within the box, 10th and 90th percentiles as whiskers and outliers as individual data points, n = 10; scale bar 10 μm.* **F)** Classification performance is insensitive to the numbers of supervoxel (SV) categories but sensitive to the number of megavoxel (MV) categories for varying SV and MV sizes. Classification performance measured as Cohen's Kappa to distinguish between neurons treated with 25 μM glutamate or incubated in BSS0 (buffer). For this dataset there is little difference between SV size 3x3x3 and MV size 3x3x1 and SV 6x6x3 and MV 6x6x1 consistent with the data in panels D-E showing that small structures do not usefully contribute to distinguishing the two treatments. However, increasing the number of MV categories improved performance from Cohen's kappa of 0.6 to 0.8. Each panel shows Cohen's kappa over 10-fold cross-validation for the different parameter conditions. Graphs display n = 10 ± SD.

**Fig D. Generation of oncogene-expressing MCF10A cell lines and organoids. A)** Western blot of the generated MCF10A cell lines confirming transgene expression. For Myc, Myc[T58A], Bcl-2, Bcl-XL, p53[R248W], p53[R273H], and SNAIL, we confirmed transgene expression using an antibody against the FLAG epitope. For PIK3CA[H1047R]-ovexpressing MCF10A cells, we confirmed transgene expression using an antibody against the phosphorylated form of the PIK3 substrate AKT (pAKT (S473)). The blot was stripped and probed for pAKT (S473) and as controls actin and total AKT. **B)** Sample images of MCF10A organoids expressing different

 

oncogenes indicated to the left). For each oncogene, the bottom, middle and top slices for a single organoid are shown. Images are from the stack of confocal micrographs pseudocolored as; *Blue–nuclei (DRAQ5), green–lysosomes (Lysotracker), red–mitochondria (Mitotracker). Scale bar: 100 μm.*

**Fig E. 2D intensity- and morphology-based features for each cluster to aid interpretation of Phindr3D feature based clustering. A-C)** Maximum projection based average pixel intensity values for the organoids stained with **A)** Lysotracker, **B)** Mitotracker, and **C)** DRAQ5 (Nucleus Intensity) for each cluster. **D)** Hollowness parameter for the different clusters. The hollowness was computed by using the center slice for each organoid and computing the ratio between DRAQ5 intensities of outer ring region (width 20 pixels) and the inner region (that excludes the ring region). A normal acinar organoid is expected to have a high value since the center is lower in intensity. However, there was very little variation in this parameter across the clusters. The reasons for this are not clear. **E)** 2D areas computed from 2D binary masks, which corresponds to the conventional way of analyzing this type of data. **F)** Number of megavoxels per acinar organoid as a surrogate marker for 3D volume. For this parameter only cluster 7 is significantly different from all other clusters. *Box plots show median (red line) and $25^{th}$ and $75^{th}$ percentiles within the box, $10^{th}$ and $90^{th}$ percentiles as whiskers and outliers as individual data points. All p-values for group-wise comparison using oneway ANOVA and Tukey-Kramer post-hoc test are shown in Table B.*

**Fig F. Deep learning analysis of oncogene-expressing organoid images.** After maximum intensity projection of the image data, a convolutional neural network (CNN) was trained using 500 of the 1330 MCF10A organoid image stacks across the 10 oncogene classes. 30 images per class were chosen for validation. **A)** t-SNE mapping of the embeddings of the CCN highlighting the heterogeneity of the resulting phenotypes. The coloring corresponds to the different genotypes. **B)** Clustering control and oncogene-expressing organoids using the seven phenotypes identified by Phindr3D (Fig 6C and 6D) demonstrated that the clustering after deep learning analysis was in good agreement to the clustering after Phindr3D analysis for many (e.g. Empty Vector, Parental, Myc, $Myc^{T58A}$, $PIK3CA^{H1047R}$), but not all genotypes. **C)** The mutual information of this clustering was calculated with 0.526 and is comparable with the MI generated with Phindr3D (0.516, Fig 6H).

**Table A: Composition of clusters of oncogene-expressing MCF10 organoids. A)** Distribution of the number of oncogene-expressing organoids across the seven clusters. **B)** Distribution of the percentage of oncogene-expressing organoids across the seven clusters. The fractions were calculated from the number of organoids for each cell line (panel A).

**Table B: Tables of p-values (Pval) for Fig E of S1 Text.** Tukey-Kramer post-hoc test performed after oneway ANOVA on the indicated data from Fig E. P-values < 0.05 are highlighted in bold.

(PDF)

## Acknowledgments

The authors would like to thank Alexander Schreiner, Karin Boettcher, and Stefan Letzsch (PerkinElmer, Hamburg, Germany) as well as Alex Kalyuzhny and Jodi Hagen (R&D Systems, Minneapolis, USA) for providing the 3D autophagy data set.

## Author Contributions

**Conceptualization:** Philipp Mergenthaler, Santosh Hariharan, David W. Andrews.

**Data curation:** Philipp Mergenthaler, Santosh Hariharan.

   

**Formal analysis:** Philipp Mergenthaler, Santosh Hariharan, James M. Pemberton, Corey Lourenco, Linda Z. Penn, David W. Andrews.

**Funding acquisition:** Philipp Mergenthaler, Linda Z. Penn, David W. Andrews.

**Investigation:** Philipp Mergenthaler, Santosh Hariharan, James M. Pemberton, Corey Lourenco.

**Methodology:** Philipp Mergenthaler, Santosh Hariharan, David W. Andrews.

**Supervision:** Philipp Mergenthaler, Santosh Hariharan, Linda Z. Penn, David W. Andrews.

**Writing – original draft:** Philipp Mergenthaler, Santosh Hariharan, James M. Pemberton, Corey Lourenco, Linda Z. Penn, David W. Andrews.

**Writing – review & editing:** Philipp Mergenthaler, Santosh Hariharan, James M. Pemberton, Corey Lourenco, Linda Z. Penn, David W. Andrews.

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
