## [Decision Letter · Decision Letter 0]

13 Aug 2020

Dear Dr. Andrews,

Thank you very much for submitting your manuscript "Rapid 3D phenotypic analysis of neurons and organoids using data-driven segmentation-free machine learning" for consideration at PLOS Computational Biology.

As with all papers reviewed by the journal, your manuscript was reviewed by members of the editorial board and by several independent reviewers. In light of the reviews (below this email), we would like to invite the resubmission of a significantly-revised version that takes into account the reviewers' comments.

All three reviewers had positive comments about the manuscript and saw potential, particularly in providing this 3D phenotyping software package in a way that is accessible to much of the biological community. However there were some major critiques that will need to be addressed, including claims about the novelty of the underlying supervoxel clustering method, and more comprehensive comparison of this method to other methods in Figure 5. Advantages and weaknesses need to be better described. Also the proposed method should be thoroughly compared, at least in the text, to deep learning methods.

We cannot make any decision about publication until we have seen the revised manuscript and your response to the reviewers' comments. Your revised manuscript is also likely to be sent to reviewers for further evaluation.

Sincerely,

Jeffrey J. Saucerman

Associate Editor

PLOS Computational Biology

Jason Haugh

Deputy Editor

PLOS Computational Biology

Reviewer's Responses to Questions

**Comments to the Authors:**

Reviewer #1: The paper presents a segmentation-free voxel classification approach to phenotypic profiling in 3D microscopy data. While the general idea of recursive supervoxel clustering for image classification is not new, and actually constitutes a standard approach in computer vision (multi-resolution pyramids), the application here seems significant and the work is of high practical value for users, in particular because a software is made available.

Suggestions for improvement:

1) The basic motivation should be toned down a bit. It is not true that phenotypic profiling of “three-dimensional microscopy datasets has been impractical”. It has actually been done before. It may not be easy, though.

2) It may be good to state the amount of training data needed for “shallow learning” right at the beginning of the results section in order to justify the use of the term “shallow learning”, which has a well-defined meaning in the machine-learning community.

3) In line 107, I would appreciate some information on how the binning is done. Based on which pixel property is the data binned? How many bins?

4) The observation that adding Gaussian noise to the data first improves performance is surprising. Could the authors elaborate on why this is? Is the classifier partially picking up noise signatures? Would performance on perfectly noise-free (synthetic) images be worse?

5) I cannot follow the claim made in the Discussion section that the method used here to determine the number of clusters is novel. Nor do I believe that it is optimal (with respect to which loss function?). It seems that the authors are using the classic L-curve method for determining the number of clusters. This method is known to be a heuristic without any optimality guarantees on the test data, and it is well known. This claim should be removed, or it should be explained why and how the methods differ.

6) The discussion section would benefit from being a bit more self-critical and also highlight the limitations of the method, or cases where such analysis would fail altogether. Clearly, the size of the supervoxels and megavoxels sets a resolution limit. I can also not imagine this method to be useful to classify phenotypes that mainly differ in morphology or shape. Nor would the method be able to distinguish different textures with the same intensity frequencies. It would be good if the authors could provide some guidelines for when someone should consider their method, to reduce the risk of user frustration.

7) From a technical point of view, I would not call the method presented here “segmentation-free”. In fact, one of the first steps in the proposed processing pipeline is to apply adaptive thresholding to the images in order to distinguish foreground pixels from background pixels. This *is* segmentation. It may not be instance segmentation, but it is the classic definition of image segmentation, and thresholding is known to be the simplest possible segmentation algorithm. In this sense, the method presented is not segmentation-free, and one may even ask the question whether it could be improved by using better (i.e., information-optimal) initial segmentation methods. This should be discussed and/or the language corrected.

In essence, what I think is done here, is weakly-supervised shallow-learning of intensity histograms in a 3-level multi-resolution pyramid over pre-segmented images. Nothing of this is new by itself, but the presented pipeline is practically useful.

Reviewer #2: The main contribution of this manuscript is a novel, super-pixel-based method that circumvents the need to explicitly segment structures and pre-define quantitative features when analyzing 3D micrographs of organoids.

The methodology is described well but the use of fixed pixel sizes to create super-voxels and mega-voxels will make the method dependent both on the magnification of the image and the size of the organoids. A similar problem exists for the method of focus stacking. The authors argue that their method is superior to deep learning methods (DLMs) because of lower computational resource needs. However, this argument is questionable because DLMs are becoming state of the art and the cost of their computational resources are well within the reach of scientific researchers and in some cases are free. Moreover, DLMs might significantly outperform the proposed method for the same tasks, in which case a DLM would be worth it even if it costs more. Therefore, the authors need to compare their method with a state of the art DLM. This is one of the two major shortcomings of the study. There needs to be some discussion about the choice of principal component analysis and Sammon mapping versus tSNE and UMAPP for projecting the data. The claim based off figures 5D, 5E and 5F that Phindr3D is better than the others is not apparent. The improvement from using Phindr3D needs to be quantitatively compared to the other methods and shown to be statistically significant.

The biological results are quite impressive and interesting, but they only confirm what was anticipated. The value of the proposed methodology can only be proven by demonstrating that the method produces new biological insights that could not have been obtained from using other, existing methods that are approximately equivalent in terms of cost (human effort, computer hardware costs, reagent costs etc). This is the second major shortcoming of the manuscript.

Reviewer #3: The manuscript of Mergenthaler et al introduces a software tool termed Phindr3D for automatic extraction of features from 3D microscopy image stacks. The software targets the problem of whole-image (in this case, whole-3D-image) classification and clustering, with classification and clustering methods based on the extracted features also provided in the software. The algorithm is applied to several biological problems where it produces results consistent with the prior knowledge.

The work has a lot of positive sides. All the data will be available at IDR, the software is provided (but I couldn't find a way to run it on Mac or Linux without a Matlab license which I believe most potential Phindr3D users are not going to possess), application is demonstrated on several very different phenotyping problems, the method itself is fast and scales well.

My main problem with the manuscript is that it presents itself as a novel computer vision method. Image classification based on spatial pyramids is a very well studied area. The seminal work of Lazebnik et al from CVPR2006 (Beyond Bag of Features: Spatial Pyramid Matching for Recognizing Natural Scene Categories) has been cited around 10,000 times, many of these citations improving the original method which is already quite close to what the authors are proposing. This whole sub-field, along with other explicit image feature methods, has largely been replaced by CNN-based classification, which, at least on natural images, has been shown too be vastly superior. The authors choose not pursue the deep learning route, as they want to run their software on regular desktop PCs without GPUs (this is how I understand their reasoning), but all this existing work needs to be acknowledged thoroughly. One small difference is that the authors operate in 3D rather than 2D, but since the binning they perform is regular this extension is straight-forward.

I am not a biologist and can not comment on the biological applications, I hope other reviewers can address these points.

**Have all data underlying the figures and results presented in the manuscript been provided?**

Reviewer #1: Yes

Reviewer #2: **No: **The manuscript clearly states, "All 3D microscopy data sets generated in this study WILL BE deposited ...."

Reviewer #3: Yes

PLOS authors have the option to publish the peer review history of their article (what does this mean?). If published, this will include your full peer review and any attached files.

Reviewer #1: No

Reviewer #2: No

Reviewer #3: No
---

## [Decision Letter · Decision Letter 1]

7 Dec 2020

Dear Dr. Andrews,

We are pleased to inform you that your manuscript 'Rapid 3D phenotypic analysis of neurons and organoids using data-driven cell segmentation-free machine learning' has been provisionally accepted for publication in PLOS Computational Biology.

We ask that you please consider all of the last comments of the Reviewers, copied below, as you prepare your final manuscript.

Best regards,

Jeffrey J. Saucerman

Associate Editor

PLOS Computational Biology

Jason Haugh

Deputy Editor

PLOS Computational Biology

Reviewer's Responses to Questions

**Comments to the Authors:**

Reviewer #1: The revisions made by the authors sufficiently addressed my concerns, and I recommend the paper for publication now.

Reviewer #2: There are a number of specific comments about this manuscript:

1) I disagree with the claim of unexpectedness in the sentence, "However, since all of the inhibitors of anti-apoptosis proteins trigger cell death by apoptosis it was unexpected that the different drugs resulted in well separated clusters." The inhibitors are not the same, apoptosis is complex engaging a variety of pathways and the inhibitors will certainly have off-target effects, some undiscovered. Therefore it is not surprising at all that different drugs resulted in well separated clusters.

2) Why is the text "For Sammon plot (B,C): ...." in italics?

Reviewer #3: The paper has improved substantially, I especially appreciate the authors toning their novelty claims down and trying out a deep learning-based solution. The limitations discussion is also a great addition which puts the whole manuscript in a much more practical setting. The choice of technology is, of course, up to the authors, but I remain worried about the potential uptake of the tool in the field if it requires not only a Matlab license, but also 2 toolboxes. Would it not be possible to create a stand-alone solution, such as the one proposed in https://www.nature.com/articles/s41592-020-0938-1? Otherwise, I have no further requests.

**Have all data underlying the figures and results presented in the manuscript been provided?**

Reviewer #1: Yes

Reviewer #2: Yes

Reviewer #3: Yes

PLOS authors have the option to publish the peer review history of their article (what does this mean?). If published, this will include your full peer review and any attached files.

Reviewer #1: **Yes: **Ivo Sbalzarini

Reviewer #2: No

Reviewer #3: No

---

## [Editor Report · Acceptance letter]

11 Feb 2021

PCOMPBIOL-D-20-01013R1 

Rapid 3D phenotypic analysis of neurons and organoids using data-driven cell segmentation-free machine learning

Dear Dr Andrews,

I am pleased to inform you that your manuscript has been formally accepted for publication in PLOS Computational Biology. Your manuscript is now with our production department and you will be notified of the publication date in due course.

With kind regards,

Alice Ellingham
